# MixNAM: Advancing Neural Additive Models with Mixture of Experts

## Abstract

Additive models, such as Neural Additive Models (NAMs), are recognized for their transparency, providing clear insights into the impact of individual features on outcomes. However, they traditionally rely on point estimations and are constrained by their additive nature, limiting their ability to capture the complexity and variability inherent in real-world data. This variability often presents as different influences from the same feature value in various samples, adding complexity to prediction models. To address these limitations, we introduce MixNAM, an innovative framework that enriches NAMs by integrating a mixture of experts, where each expert encodes a different aspect of this variability in predictions from each feature. This integration allows MixNAM to capture the variability in feature contributions through comprehensive distribution estimations and to include feature interactions during expert routing, thus significantly boosting performance. Our empirical evaluation demonstrates that MixNAM surpasses traditional additive models in performance and is comparable to complex black-box approaches. Additionally, it improves the depth and comprehensiveness of feature attribution, setting a new benchmark for balancing interpretability with performance in machine learning. Moreover, the flexibility in MixNAM configuration facilitates the navigation of its trade-offs between accuracy and interpretability, enhancing adaptability to various data scenarios.

## 1 Introduction

Deep neural networks (DNNs) have proven exceptional at modeling complex data relationships, achieving remarkable results across various domains, such as computer vision and natural language processing (Young et al., 2018; Hassaballah & Awad, 2020). Despite these successes, the opaque nature of DNN outputs often makes them non-interpretable, which hinders their usage in high-stakes areas like healthcare and finance where the basis of decisions is crucial (Ghnemat et al., 2023; Liao & Varshney, 2021). This critical limitation has spurred research into interpretable machine-learning models (Alvarez Melis & Jaakkola, 2018; Ghorbani et al., 2019; Wu et al., 2020; Yeh et al., 2020). These models aim to make the reasoning behind decisions more transparent, facilitating trust and wide adoption in sensitive applications.

Among the interpretable approaches developed, additive models stand out for their transparency, enabling direct visualization of how individual features impact predictions (Hastie, 2017). Additive models independently encode each feature through a non-linear transformation and subsequently combine them linearly for final predictions, simplifying the understanding of the influence from each feature. Based on the framework of additive models, Neural Additive Models (NAMs; Agarwal et al., 2021) attempt to model the complex patterns of each feature with DNNs, offering a flexible and differentiable representation of features while maintaining the additive structure.

While the additive nature of feature information in NAMs allows for clear explanations of individual features, this structure inherently limits their ability to learn intricate mappings between input features and target outputs. Thus, the performance of these models often lags behind that of their complex black-box counterparts, which can learn interactions among features. Furthermore, additive models traditionally produce a single outcome for each feature value, an approach that fails to account for the variability in outcomes that occur in complex real-world scenarios. The variability in model prediction refers to the extent to which the predicted outputs of a model fluctuate or differ

when the same input feature value is presented across different samples. For example, the same health metric might have different implications depending on an individual's age group or other contextual factors. By relying on point-to-point estimations, these models do not capture the actual distribution of possible outcomes associated with a given feature value. Recent attempts to enhance NAMs with uncertainty estimation seek to address this issue (Bouchiat et al., 2023; Thielmann et al., 2024). However, these approaches are still limited by their inevitable assumptions of prior distributions and the additive nature of the entire models, which restrict their ability to accurately reflect complex underlying distributions and interactions.

To address the limitations of traditional additive models while retaining their interpretative benefits, we introduce MixNAM, a novel and general framework that extends beyond the constraints of NAMs. MixNAM integrates multiple expert feature encoders operating in parallel, each of which encodes a different aspect of the variability in model predictions. Moreover, a dynamic routing mechanism is introduced to assess and combine the relevance of different experts for the final prediction. Our experiments demonstrate that MixNAM significantly outperforms all existing additive models, matching the performance of complex black-box methods. Different from traditional additive models that offer point estimations, MixNAM provides a more nuanced understanding of feature impacts by visualizing the range and distribution of possible outcomes for each feature value. The results further reveal that MixNAM can be seen as a generalization of NAM, with the flexibility to adjust the balance between detailed distribution estimations and straightforward point estimations, catering to specific needs for transparency and performance. The contributions of this work can be summarized as follows:

- We propose MixNAM, a general framework built upon NAMs, by incorporating a mixture of experts with dynamic routing algorithms.

- MixNAM demonstrates superior performance compared to traditional additive models and is comparable to complex black-box models.

- MixNAM preserves the capability of explaining feature contributions through visualizations, enhancing the current point estimations with nuanced distribution estimations.

- MixNAM enables a seamless transition from detailed distribution estimation to precise point estimation, effectively balancing model accuracy and interpretability.

## 2 BACKGROUND

### 2.1 GENERALIZED ADDITIVE MODELS

Additive models, or Generalized Additive Models (GAMs; Hastie, 2017), represent an evolving area in the field of explainable artificial intelligence, blending interpretability with predictive accuracy. Given a sample with $n$ features $x_1, \cdots, x_n$, the task of an additive model, in general, is to learn corresponding feature encoders $f_1, \cdots, f_n$ which map the input values from feature domains to the prediction domain. The final predicted value of the target output $y$ is computed as

$$\hat{y} = w_0 + \sum_{i=1}^{n} f_i(x_i),\tag{1}$$

where $w_0$ is the learnable bias of the output distribution. GAMs highlight the influence of individual features additively, facilitating easy comprehension and interpretation. Advances such as Explainable Boosting Machine (EBM; Lou et al., 2012) and Neural Oblivious Decision Ensembles for GAM (NODE-GAM; Chang et al., 2022) have built upon this foundation by integrating boosting techniques (Friedman, 2001) and differentiable decision trees (Popov et al., 2020), respectively, enhancing accuracy while preserving interpretability.

### 2.2 NEURAL ADDITIVE MODELS

Developed from traditional additive models, Neural Additive Models (NAMs; Agarwal et al., 2021) incorporate neural networks to encode each feature, marking a significant evolution in the modeling capability of GAMs. This neural integration allows NAMs to capture more complex feature relationships without sacrificing the interpretative benefits. Subsequent developments, such as Neural Basis

Models (NBMs; Radenovic et al., 2022) and Gaussian Process Neural Additive Models (GP-NAMs; Zhang et al., 2024) have further refined this approach. They focus on efficiency and scalability, reducing the complexity and enhancing the ability of models to manage larger datasets with complex underlying structures. Recent research has also explored extending NAMs to include uncertainty estimation, providing a more comprehensive explanation of feature influence by considering distribution parameters and Bayesian inference methods (Bouchiat et al., 2023; Thielmann et al., 2024). However, these models generally rely on a simplistic assumption about output distributions, which may limit their effectiveness in real-world scenarios where data interactions are complex.

## 2.3 MIXTURE OF EXPERTS

The Mixture of Experts (MoE) technique has gained substantial attention from the research community recently for its ability to enhance both the scalability and expertise of neural networks. It is designed to learn multiple specialized sub-models, known as "experts", on different subtasks. A gating network is trained to dynamically assign a new input to the most relevant experts (Jacobs et al., 1991). One key advantage of the MoE model is its efficiency and adaptability in handling large-scale problems. Shazeer et al. (2017) explored scaling MoE for applications in massive neural network architectures by incorporating a sparse gating network to select only a small subset of experts for each input. They demonstrate that integrating MoE layers can significantly boost model capacity with a minor loss in computational efficiency. Recent work also explores the adaptation of MoE in the Transformer architecture (Vaswani et al., 2017), encouraging their use in cutting-edge research such as large language models (Artetxe et al., 2022; Jiang et al., 2024; Lepikhin et al., 2021) and vision language models (Lin et al., 2024; Shen et al., 2023).

## 3 MIXNAM: MIXTURE OF NEURAL ADDITIVE MODELS

Our Mixture of Neural Additive Models (MixNAM) framework extends traditional Neural Additive Models (NAMs) by incorporating multiple expert predictors for each feature to better capture the variability and complexity of data relationships. This approach combines feature-specific expert encoders with a dynamic expert routing mechanism to determine the most relevant experts for each input scenario, as depicted in Figure 1.

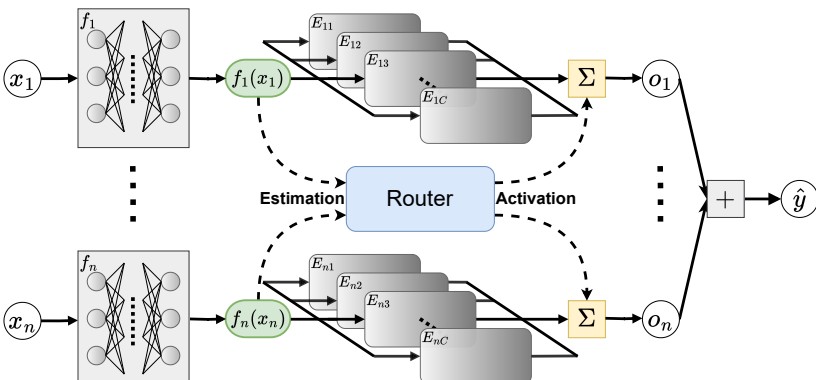

Figure 1: Architecure of our Mixture of Neural Additive Models (MixNAM) framework.

## 3.1 FEATURE ENCODING WITH EXPERTS

In traditional additive models, a single predictor associated with each feature often limits the expressiveness and adaptability of the model in complex scenarios. MixNAM addresses this by embedding a mixture of $C$ expert predictors for each feature $x_i$ ($i \in \{1, \cdots, n\}$), significantly enhancing the model's capacity to capture diverse non-linear relationships within the data. Specifically, each feature $x_i$ is encoded by an expressive feature encoder $f_i$, which transforms the input into a latent vector. We then enhance the standard encoding process by introducing $C$ expert predictors $E_{i1}, \cdots, E_{iC}$, where each predicts different potential outcomes from the encoded information of $x_i$:

$$o_{ik} = E_{ik}(f_i(x_i)), \quad \forall i \in \{1, \cdots, n\}, k \in \{1, \cdots, C\}. \tag{2}$$

This architecture allows each feature to express a spectrum of influences, capturing subtle variations in the data that might be overlooked by more monolithic approaches. In our experiments, each $f_i$ is implemented as a multi-layer neural network, and $E_{ik}$ is implemented as a linear layer.

## 3.2 DYNAMIC EXPERT ROUTING

Expert routing is pivotal in MixNAM, where the relevance of each expert is dynamically assessed through a routing mechanism. MixNAM uses a router to compute relevance scores for experts based on the overall context of input and aggregates expert predictions for final outputs:

$$o_i = \sum_{k=1}^{C} r_{ik}o_{ik} \quad \text{w.r.t.} \quad r_{ik} \geq 0 \text{ and } \sum_{k=1}^{C} r_{ik} = 1, \quad \forall i \in \{1, \cdots, n\}, \tag{3}$$

where $r_{ik}$ is the estimated relevance for the $k$-th expert in feature $x_i$, and $o_i$ is the summarized outcome given by $x_i$. The final prediction $\hat{y}$ is generated by adding outputs $o_1, o_2, \cdots, o_n$ from each feature with an additional bias term $\omega_0$ similar to Formula 1.

**Score Estimation.** Motivated by the fact that one specific feature input may correspond to different target values depending on the overall information of the instance, we propose a routing mechanism that takes all features as input and estimates the relevance of each expert for prediction. Suppose the encoded information $f_i(x_i)$ from $f_i$ is represented by a $d$-dimensional vector. For the $C$ experts of the feature $x_j$, the routing system in MixNAM learns a $d \times C$ scoring matrix $\mathcal{A}_{ij}$ to assess how information encoded from $x_i$ affects the expert selection of $x_j$. The summarized scores for expert relevance estimation in feature $x_j$ can be produced by aggregating the information from all features:

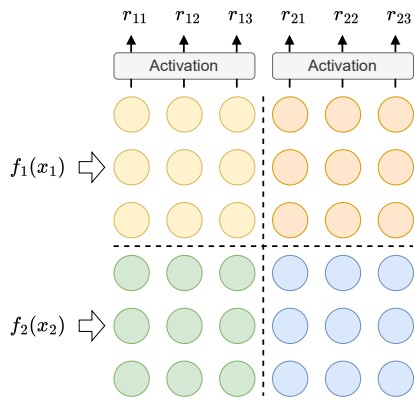

$$\varphi_j = \mu_j + \sum_{i=1}^{n} \mathcal{A}_{ij}^{\top} f_i(x_i) \tag{4}$$

where $\mu_j$ is a learnable $C$-dimensional bias term. The scores in $\varphi_j$ are then normalized into a valid relevance distribution over experts using an activation strategy.

Figure 2: An illustration of our routing mechanism. This example contains two features with three experts each.

As depicted in Figure 2, all scoring matrices can be organized as a large block matrix, where the block at the $i$-th row and the $j$-th column corresponds to $\mathcal{A}_{ij}$. Leveraging parallel computing resources, such as GPUs, this organization facilitates accelerated score estimation by processing multiple features parallelly. We also explore a specialized version of MixNAM in Appendix G, where the scoring matrices compose a block diagonal matrix, making it a strictly additive model.

**Activation Strategy.** To effectively transform the raw scores into a valid distribution of expert relevance, an activation strategy is needed to project $\varphi_i \in \mathbb{R}^C$ to the simplex $\mathcal{D} = \{v \in \mathbb{R}_+^C | v^\top 1 = 1\}$. This ensures that the relevance scores fulfill the constraints required by the model, where each score vector must sum to one and contain only non-negative values.

We utilize a sparse activation strategy adapted from previous MoE research (Jiang et al., 2024) to focus on the most significant experts without overwhelming the model with redundant information. Given scoring vector $\varphi_i$, our model will first create a masking vector $M_i \in \mathbb{R}^C$ to zero out less significant scores, whose $k$-th entry is defined as

$$M_i[k] = \begin{cases} 0, & \text{if } \varphi_i[k] \text{ is among the largest } K \text{ values in } \varphi_i, \\ -\infty, & \text{otherwise,} \end{cases} \tag{5}$$

where $K$ is the number of experts activated. The relevance scores will then be computed by

$$r_{ik} = \exp(\varphi_i[k] + M_i[k]) \Big/ \sum_{l=1}^{C} \exp(\varphi_i[l] + M_i[l]). \tag{6}$$

Such an activation strategy results in a continuous subset of values in $\mathcal{D}$, allowing for a nuanced learning of expert relevance. While the router is flexible for learning a continuous range of outputs, our analysis in Appendix F proves that it is not a universal function approximator but is inherently a generalized additive model with normalization. In Appendix H, we also discuss an alternative activation strategy that brings a finite set of points in $\mathcal{D}$ as a discrete range for relevance activation. It shows distinct model behaviors that the MixNAM framework achieves with different configurations.

### 3.3 OPTIMIZATION AND TRAINING

The training of MixNAM involves optimizing standard loss functions adapted to downstream tasks, such as cross-entropy or mean squared error. Following previous studies (Agarwal et al., 2021; Radenovic et al., 2022), we employ the dropout and L2 regularization on model parameters to avoid overfitting and promote robust generalization. An L2 penalty of feature outcomes is also applied to suppress redundancy. Additionally, we introduce an expert dropout to prevent the model from overly depending on specific routing paths. An expert variation penalty is also introduced, designed to minimize variance in predictions across experts for the same input feature, encouraging the model to learn meaningful and interpretable expert functions. This penalty is the key to balance accuracy and interpretability in MixNAM, which is discussed in detail in Section 4.3.

As introduced in Section 3.2, the final prediction of an input sample by MixNAM is obtained by linearly aggregating the predicted outcomes from each feature

$$\hat{y} = \omega_0 + \sum_{i=1}^{n} o_i = \omega_0 + \sum_{i=1}^{n} \sum_{k=1}^{C} r_{ik} o_{ik}. \tag{7}$$

Suppose we have $N$ training samples $x^1, \cdots, x^N$ with corresponding task labels $y^1, \cdots, y^N$. The objective during training is

$$\text{minimize} \quad \frac{1}{N} \sum_{t=1}^{N} \mathcal{L}(y^t, \hat{y}^t) + \frac{\gamma}{nN} \sum_{t=1}^{N} \sum_{i=1}^{n} (o_i^t)^2 + \frac{\lambda}{nNC} \sum_{t=1}^{N} \sum_{i=1}^{n} \sum_{k=1}^{C} \left( o_{ij} - \frac{o_{i1} + \cdots + o_{iC}}{C} \right)^2,$$

$$\tag{8}$$

where $\hat{y}^t$ denotes the final model prediction for $x^t$ ($t \in \{1, \cdots, N\}$). $\gamma$ is the weight for the output penalty and $\lambda$ is the weight for the expert variation penalty. The loss function $\mathcal{L}$ can be the mean square error loss or the cross-entropy loss, depending on the task that the model is dealing with.

## 4 EXPERIMENTS

The experiments in this section are designed to comprehensively evaluate MixNAM across dimensions of accuracy and interpretability, core attributes that define its utility in practical applications. We begin by assessing the accuracy of MixNAM through performance comparisons with established baselines ranging from traditional additive models to complex black-box models. We then delve into the interpretability aspect, utilizing visualizations to demonstrate how MixNAM elucidates the impact of individual features on predictions. Moreover, experiments are conducted to showcase the unique ability of MixNAM to balance accuracy with interpretability, illustrating how this can be adjusted to meet specific application demands. In addition to the experiments on real-world datasets, we test our MixNAM with simulated data to see if the MoE design helps it capture multimodal data. Additional results in Appendix I reveal how the performance of MixNAM scales with the number of experts in its configuration.

### 4.1 EVALUATION OF MIXNAM ACCURACY

#### 4.1.1 DATASETS

To evaluate the accuracy of MixNAM, we select six widely recognized datasets that span a mix of regression and classification tasks: Housing (Pace & Barry, 1997), MIMIC-II (Saeed et al., 2011), MIMIC-III (Johnson et al., 2016), Income (Blake, 1998), Credit, and Year. These datasets are chosen due to their varying complexities, including differences in the number of instances, features, and the presence of categorical variables, ensuring a comprehensive assessment of the model performance across diverse data scenarios. Detailed information about each dataset can be found in Appendix A.

### 4.1.2 BASELINES

To evaluate the accuracy of MixNAM, we benchmark it against a range of models selected for their relevance to the goals of MixNAM. This includes linear and spline models, which provide basic feature explanation capabilities similar to additive models, Neural Additive Models (NAMs; Agarwal et al., 2021) and Neural Basis Models (NBMs; Radenovic et al., 2022), which use neural networks to enhance feature encoding, Explainable Boosting Machine (EBM; Lou et al., 2012; Nori et al., 2019) and NODE-GAM (Chang et al., 2022), which integrate advanced techniques like gradient boosting (Friedman, 2001) and differentiable decision trees (Popov et al., 2020). Additionally, extended additive models such as EB$^2$M, NA$^2$M, NB$^2$M, and NODE-GA$^2$M, which encode pairwise feature interactions, are assessed. While these models facilitate interaction-based explanations, they lack the capacity to succinctly explain the impact of individual features, presenting challenges in attribute summarization due to potential dimensionality issues. Traditional black-box models like MLP, NODE (Popov et al., 2020), and XGBoost (Chen & Guestrin, 2016) are also evaluated for their ability to flexibly encode feature interactions, though at the cost of interpretability. The implementation details of our experiments are presented in Appendix B and C.

### 4.1.3 EVALUATION RESULTS

Table 1 presents the comparison of MixNAM against all baseline models. Evaluation metrics are provided below each dataset name, with arrows indicating the desired direction for scores. For each dataset, the top four scores are underlined to highlight a broad range of competitive models. In addition to the evaluation scores, a column labeled "FA" (Feature Attribution) is added to the table, indicating whether the method can provide explanations focused on individual feature impacts.

Table 1: Performance comparison on benchmark datasets. "FA" (Feature Attribution) capabilities of different models are denoted by ✔ for presence and ✘ for absence. The top four scores for each dataset are underlined. MixNAM significantly outperforms traditional additive models with "FA".

| Model | FA | Housing RMSE ↓ | MIMIC-II AUC ↑ | MIMIC-III AUC ↑ | Income AUC ↑ | Credit AUC ↑ | Year MSE ↓ |
|---|---|---|---|---|---|---|---|
| MLP | ✘ | 0.501 ± 0.006 | 0.835 ± 0.014 | 0.815 ± 0.009 | 0.914 ± 0.003 | 0.981 ± 0.007 | 78.48 ± 0.56 |
| NODE | ✘ | 0.523 ± 0.000 | 0.843 ± 0.011 | 0.828 ± 0.007 | 0.919 ± 0.003 | 0.981 ± 0.009 | 76.21 ± 0.12 |
| XGBoost | ✘ | 0.443 ± 0.000 | 0.844 ± 0.012 | 0.819 ± 0.004 | 0.928 ± 0.003 | 0.978 ± 0.009 | 78.53 ± 0.09 |
| EB$^2$M | ✘ | 0.492 ± 0.000 | 0.848 ± 0.012 | 0.821 ± 0.004 | 0.928 ± 0.003 | 0.982 ± 0.006 | 83.16 ± 0.01 |
| NA$^2$M | ✘ | 0.492 ± 0.008 | 0.843 ± 0.012 | 0.825 ± 0.006 | 0.912 ± 0.003 | 0.985 ± 0.007 | 79.80 ± 0.05 |
| NB$^2$M | ✘ | 0.478 ± 0.002 | 0.848 ± 0.012 | 0.819 ± 0.010 | 0.917 ± 0.003 | 0.978 ± 0.007 | 79.01 ± 0.03 |
| NODE-GA$^2$M | ✘ | 0.476 ± 0.007 | 0.846 ± 0.011 | 0.822 ± 0.007 | 0.923 ± 0.003 | 0.986 ± 0.010 | 79.57 ± 0.12 |
| Linear | ✔ | 0.735 ± 0.000 | 0.796 ± 0.012 | 0.772 ± 0.009 | 0.900 ± 0.002 | 0.976 ± 0.012 | 88.89 ± 0.40 |
| Spline | ✔ | 0.568 ± 0.000 | 0.825 ± 0.011 | 0.812 ± 0.004 | 0.918 ± 0.003 | 0.982 ± 0.011 | 85.96 ± 0.07 |
| EBM | ✔ | 0.559 ± 0.000 | 0.835 ± 0.011 | 0.809 ± 0.004 | 0.927 ± 0.003 | 0.974 ± 0.009 | 85.81 ± 0.11 |
| NAM | ✔ | 0.572 ± 0.005 | 0.834 ± 0.013 | 0.813 ± 0.003 | 0.910 ± 0.003 | 0.977 ± 0.015 | 85.25 ± 0.01 |
| NBM | ✔ | 0.564 ± 0.001 | 0.833 ± 0.013 | 0.806 ± 0.003 | 0.918 ± 0.003 | 0.981 ± 0.007 | 85.10 ± 0.01 |
| NODE-GAM | ✔ | 0.558 ± 0.003 | 0.832 ± 0.011 | 0.814 ± 0.005 | 0.927 ± 0.003 | 0.981 ± 0.011 | 85.09 ± 0.01 |
| **MixNAM** | ✔ | 0.451 ± 0.002 | 0.847 ± 0.014 | 0.825 ± 0.006 | 0.927 ± 0.003 | 0.982 ± 0.009 | 78.66 ± 0.21 |

From the table, it can be observed that traditional additive models, while capable of providing feature attributions, do not generally perform as well as models that can encode feature interactions, such as NA$^2$Ms and various black-box models. However, MixNAM distinguishes itself by offering competitive performance, achieving comparable results to models encoding feature interactions, while also providing the valuable capability of feature-level explanation as indicated by the "FA" column. This notable combination of high performance and strong interpretability in MixNAM underscores its effectiveness in complex real-world applications.

## 4.2 INTERPRETABILITY OF MixNAM PREDICTION

Beyond its superior performance relative to additive models, MixNAM can explain how individual features influence the final prediction as well. MixNAM utilizes a dynamic routing mechanism where the relevance of different experts for a feature is confined within a simplex. This constraint allows MixNAM to determine precise upper and lower prediction bounds for any given feature value $x_i$ with

$$u_i = \max_{k \in \{1, \cdots, C\}} E_{ik}(f_i(x_i)) \quad \text{and} \quad l_i = \min_{k \in \{1, \cdots, C\}} E_{ik}(f_i(x_i)). \quad (9)$$

The bounds represent the maximum and minimum potential outputs for a feature, which ensure that the outputs remain within a definite and interpretable range. Additionally, by iterating over all instances, we can plot the actual feature influences after expert routing, showing how predictions distribute within the bounds.

Figure 3 illustrates the interpretative capability of MixNAM, using the "Longitude" feature from the Housing dataset as an example. For comparison, we include visual explanations by EBM and NAM on this feature. We also include the "Longitude-Latitude" interaction plot from NA$^2$M to validate whether the distribution of predictions from MixNAM aligns with models that encode complex interactions. In the figure of MixNAM, the upper bound is highlighted in dark blue, and the lower bound is in red. The actual predicted values are plotted as blue dots, showing the entire distribution of scores in the dataset. The y-axis in the figures of EBM, NAM, and MixNAM shows the mean-centered contribution to the model prediction given by different feature values. We also plot the color bars in the background to reflect the normalized data density following prior studies (Agarwal et al., 2021; Radenovic et al., 2022). In the NA$^2$M figure, the contribution given by each pair of feature values is reflected by the color of the corresponding dot.

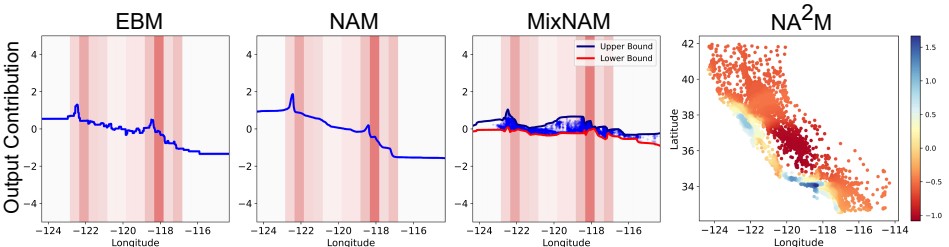

Figure 3: Visual comparison of the "Longitude" feature impact on house prices across models.

The visualizations in Figure 3 reveal that EBM, NAM, and MixNAM all capture similar geographic trends affecting house prices, with sharp increases observed around the key longitude markers at -122.5 (122.5°W, San Francisco) and -118.5 (118.5°W, Los Angeles). Compared with EBM and NAM, MixNAM provides a more comprehensive insight into the output distribution by leveraging dynamic expert routing that incorporates all features. Specifically, while high prediction scores in regions like San Francisco and Los Angeles align with other models, MixNAM captures a broad range of possible outcomes between -120 and -119, with most data points clustering around the lower bound. This detailed variance is validated by the "Longitude-Latitude" interaction analysis from NA$^2$M, where coastal regions are shown to positively impact house prices (blue dots) while other areas in the same longitude range generally lower property values (red dots). In contrast to NA$^2$M, which requires the selection of specific interactions to convey such insights, MixNAM seamlessly integrates and displays the complex underlying data distribution in a single plot, enhancing interpretative clarity and effectiveness.

### 4.3 BALANCE BETWEEN ACCURACY AND INTERPRETABILITY

In addition to demonstrating the accuracy and interpretability of MixNAM, we further explore how these two characteristics can be adjusted to reach a balance. As mentioned in Section 3.3, an expert variation penalty is implemented during training to ensure that MixNAM develops meaningful expert functions. Our qualitative and quantitative analyses demonstrate that, by adjusting the penalty weight $\lambda$, MixNAM presents a general framework of "constrained interpretable models", which provides an opportunity to control the trade-offs between model accuracy and its interpretability.

#### 4.3.1 QUALITATIVE ANALYSIS OF SHAPE PLOTS BY MIXNAM

While the default value of the variation penalty weight $\lambda$ is 0.1, we rerun our experiments on the Housing dataset with diverse weight values from 0 to 100, examining how the model performance and explanation change with different $\lambda$. Table 2 illustrates the impact of varying expert variation penalties on MixNAM. It reveals that MixNAM remains stable performance when $\lambda \leq 0.1$. However, there is a noticeable decline in model performance as $\lambda$ increases beyond the threshold.

Table 2: Performance and explanations of MixNAM with different variation penalties.

| $\lambda = 0$ | $\lambda = 0.1$ | $\lambda = 1$ | $\lambda = 10$ | $\lambda = 100$ |
|---|---|---|---|---|
| RMSE ($\downarrow$) | | | | |
| $0.451 \pm 0.003$ | $0.451 \pm 0.003$ | $0.458 \pm 0.003$ | $0.515 \pm 0.008$ | $0.582 \pm 0.002$ |
| Shape Plot | | | | |

Consistent with the stable performance for $\lambda \leq 0.1$, it is shown by the shape plots in Table 2 that the distribution of the predicted values remains similar across such settings. However, as $\lambda$ decreases further, the gap between the upper and lower bounds widens, resulting in looser bounds that become less informative and fail to accurately reflect the true limits of the data distribution. Conversely, larger $\lambda$ values tighten these bounds, better reflecting the distribution that the model captures. Nevertheless, excessively high $\lambda$ values may also constrain the ability of MixNAM to learn complex underlying data distribution, shifting the focus from estimating distributions to point estimation. The learned shape plot becomes almost identical to the ones generated by traditional additive models when the variation penalty weight is set too large (e.g., $\lambda = 100$). The varying shape plots qualitatively show how the variation penalty controls the trade-off between model accuracy and interpretability, providing different visualization results tailored for various interpretation needs.

#### 4.3.2 QUANTITATIVE ANALYSIS OF MODEL ADDITIVITY AND BOUND TIGHTNESS

In addition to the qualitative analysis of shape plots, we design two metrics, model additivity and bound tightness, to quantitatively explain how the trade-off can be controlled by the variation penalty in MixNAM. To show how much additivity is preserved in MixNAM and how close it is to a strict additive model, we look into the gradient of the model prediction with respect to each feature and decompose its variation to examine the driven factors in the variation. The total variance of the derivative of $\hat{y}$ with respect to feature $x_i$ can be decomposed as

$$\text{Var}\left(\frac{\partial \hat{y}}{\partial x_i}\right) = \mathbb{E}\left(\text{Var}\left(\frac{\partial \hat{y}}{\partial x_i}|x_i\right)\right) + \text{Var}\left(\mathbb{E}\left(\frac{\partial \hat{y}}{\partial x_i}|x_i\right)\right). \tag{10}$$

The additivity of a model on a feature is measured by quantifying how much of its derivative is determined by the feature itself, and the additivity of the entire model can be computed as:

$$\text{Additivity} = \frac{1}{n}\sum_{i=1}^{n} \frac{\text{Var}(\mathbb{E}(\frac{\partial \hat{y}}{\partial x_i}|x_i))}{\text{Var}(\frac{\partial \hat{y}}{\partial x_i})}, \tag{11}$$

where $x_1, \cdots, x_n$ are the $n$ features considered. The value of an additivity score will range from 0 to 1, where 1 means strictly additive and 0 means non-additive. In addition to the model additivity, we also examined the tightness of estimated bounds given by MixNAM in the shape plots, which is an important factor when interpreting the results. The bound tightness is measured by

$$\text{Tightness} = \frac{1}{n} \sum_{i=1}^{n} \mathbb{E}\left[ \frac{\max(o_i|x_i) - \min(o_i|x_i)}{\text{upper}(o_i|x_i) - \text{lower}(o_i|x_i)} \right], \tag{12}$$

where the $o_i$ is the predicted output given by the feature encoders of $x_i$ as defined in Formula 3. This metric measures how tightly the bounds fit actual model outputs.

Table 3 presents the model additivity and bound tightness of MixNAM with different $\lambda$ values. As the additivity metric is generalizable to all differentiable models, we include the measured additivity of MLP, NA$^2$M, and NAM for a comprehensive comparison. The results show a clear trade-off between model additivity and performance, as models with high additivity values tend to perform worse on the given task. By mitigating the additivity restrictions, MixNAM has significantly better performance with reduced additivity, which, nevertheless, is still much higher than models without feature attributions such as MLP and NA$^2$M. Moreover, the results quantitatively show that $\lambda$ can adjust the trade-off between additivity and accuracy, with model additivity close to 1 when $\lambda$ is large.

Table 3: Quantitative analysis of model additivity and bound tightness on the Housing dataset.

| | MLP | NA$^2$M | MixNAM ($\lambda=0$) | MixNAM ($\lambda=0.1$) | MixNAM ($\lambda=1$) | MixNAM ($\lambda=10$) | MixNAM ($\lambda=100$) | NAM |
|---|---|---|---|---|---|---|---|---|
| Additivity | 0.085 ±0.011 | 0.170 ±0.020 | 0.366 ±0.003 | 0.381 ±0.013 | 0.416 ±0.003 | 0.746 ±0.003 | 0.998 ±0.000 | 1.000 ±0.000 |
| Tightness | – | – | 0.860 ±0.000 | 0.953 ±0.011 | 0.988 ±0.000 | 0.999 ±0.000 | 1.000 ±0.000 | – |
| RMSE ($\downarrow$) | 0.501 ± 0.006 | 0.492 ± 0.008 | 0.451 ± 0.003 | 0.451 ± 0.003 | 0.458 ± 0.003 | 0.515 ± 0.008 | 0.582 ± 0.008 | 0.572 ± 0.005 |

Similarly, the table shows that the tightness score in MixNAM grows with the increase of the variation penalty, which already achieves 0.953 when $\lambda = 0.1$. However, a lower tightness score does not necessarily correspond to better performance, as the bounds may be too loose to reflect the actual distribution of model outputs, affecting the interpretability of shape plots given by the estimated bounds. With the bound tightness metric, we can conveniently evaluate the interpretation quality of MixNAM given by its upper and lower bounds without plotting the actual shape functions.

### 4.4 SIMULATION STUDY ON UNIMODAL AND MULTIMODAL DATA

The Mixture of Experts (MoEs) is known to be effective in capturing multi-distribution or multimodal data, which can perform better than unimodal models in certain complex data scenarios (Shazeer et al., 2017). To examine whether the MoE design in MixNAM helps it learn complex multimodal data, we perform simulation studies on MixNAM and NAM using synthetic data with known ground truth to test if MixNAM can capture the underlying patterns.

Suppose we have two random variables $x_1 \sim U(0,1)$ and $x_2 \sim Bernoulli(0.5) \times 2 - 1$. We first simulate 10k samples following a simple unimodal data distribution, where the target $y$ value is generated as

$$y = \sin(4\pi x_1) + x_2 + \varepsilon \qquad \text{with} \qquad \varepsilon \sim \mathcal{N}(0, 0.1^2) \tag{13}$$

For the multimodal data, we generate 10k samples with two modes and define $y$ as

$$y = \begin{cases} \sin(4\pi x_1) + x_2 + \varepsilon & \text{if} \quad (x_1, x_2) \in \{(x_1, x_2)|x_2 = 1\}, \\ -\sin(4\pi x_1) + x_2 + \varepsilon & \text{if} \quad (x_1, x_2) \in \{(x_1, x_2)|x_2 = -1\}. \end{cases} \tag{14}$$

Figure 4 shows the shape functions learned from the unimodal and multimodal data by NAM and MixNAM, respectively. For the unimodal data where the data-generating process is additive, both NAM and MixNAM effectively capture the relations between the features and the target output. However, on the multimodal data where the relation between $x_1$ and $y$ exhibits two distinct modes,

NAM fails to capture the complex data patterns and instead learns only a weak influence of the feature on the output. Instead, our MixNAM effectively captures the intricate relation between $x_1$ and $y$, predicting a periodic pattern in its upper and lower bounds, which aligns with the true underlying data-generating process.

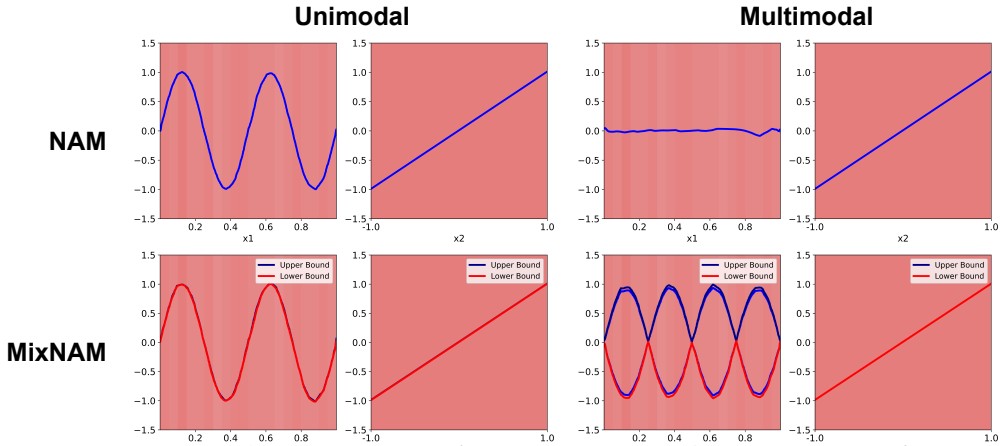

Figure 4: Shape plots of MixNAM and NAM on the simulated data.

In addition to the shape plots with estimated bounds given by MixNAM, we also visualize the individual experts learned by MixNAM in Figure 5 to see if the expert functions capture the designed modes in our simulated data. For both data distributions, we train a MixNAM model with two experts both of which can be activated. The figure shows that both learned experts on the unimodal data fit the actual data distribution and overlap with each other. It demonstrates that MixNAM can behave as a strictly additive model if the data-generating process is truly additive. On the multimodal data, the two experts in MixNAM capture similar periodic patterns but with opposite values, aligning with the two modes in the true data distribution. Our simulation study demonstrates that by incorporating MoE design into additive models, MixNAM effectively captures the complex patterns in multi-distribution and multimodal data, which the original NAM can hardly handle.

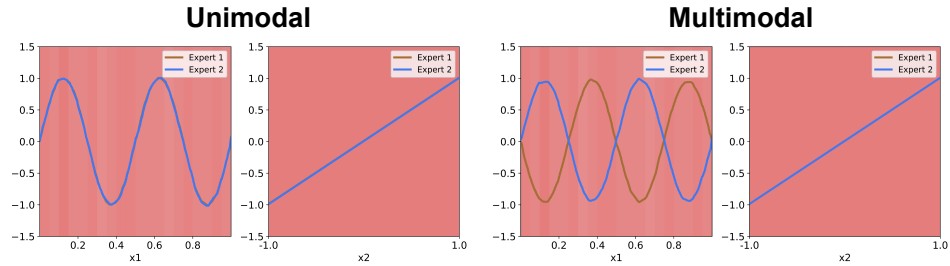

Figure 5: Learned experts in MixNAM for the simulated data.

## 5 CONCLUSION

We introduce MixNAM, a general framework that advances additive models by incorporating a mixture of experts. Our experiments show that MixNAM overcomes the inherent limitations of traditional additive models, enhancing accuracy while preserving interpretability. MixNAM not only provides detailed explanations of feature output distributions but also allows for a flexible balance between accuracy and interpretability, adapting to various application needs. This framework extends the utility of additive models, providing advanced performance and insightful interpretability, which are crucial for real-world machine-learning applications.

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

## A DATASET DESCRIPTION

**Housing Dataset.** The California Housing dataset, cited from Pace & Barry (1997), encompasses a regression task based on median housing prices across California's census blocks. It comprises 20,640 instances, each characterized by 8 distinct attributes.

**MIMIC-II Dataset.** The MIMIC-II (Multiparameter Intelligent Monitoring in Intensive Care) dataset, as described by Saeed et al. (2011), facilitates a binary classification task aimed at predicting mortality in intensive care units (ICUs). It contains 17 attributes, including 7 categorical variables.

**MIMIC-III Dataset.** MIMIC-III is a more extensive and detailed iteration of the MIMIC database referenced from Johnson et al. (2016). In our study, the same setting in NODE-GAM (Chang et al., 2022) is adopted, wherein categorical variables have been transformed into dummy variables for enhanced analysis.

**Income Dataset.** Originating from the UCI Machine Learning Repository (Blake, 1998), the Income dataset underpins a binary classification task. Its objective is to predict individuals earning in excess of $50,000 annually.

**Credit Dataset.** The Credit dataset[1] provides samples for a binary classification task on transaction fraud detection. It contains 30 anonymized features including 28 coefficients of PCA components. In Credit, 492 out of 284,807 transactions are labeled as frauds.

**Year Dataset.** The Year dataset[2] contains data for a regression task, which uses the audio features to predict the release year of a song. It includes 515,345 samples with 90 features.

The statistics of each dataset can be found in Table 4.

Table 4: Statistics of real-world datasets used in model evaluation.

| Dataset | Task | #Instances | #Features | #Categorical Features | #Classes |
|---|---|---|---|---|---|
| Housing | regression | 20,640 | 8 | 0 | – |
| MIMIC-II | classification | 24,508 | 17 | 7 | 2 |
| MIMIC-III | classification | 27,348 | 57 | 0 | 2 |
| Income | classification | 32,561 | 14 | 8 | 2 |
| Credit | classification | 284,807 | 30 | 0 | 2 |
| Year | regression | 515,345 | 90 | 0 | – |

## B IMPLEMENTATION DETAILS

For a comprehensive and equitable evaluation, we employed the officially released codes for baseline models including NODE, NODE-GAM/NODE-GA$^2$M, EBM/EB$^2$M, and NBM/NB$^2$M. We adopted the PyTorch implementation of NAM/NA$^2$M by NBM, which has been benchmarked against their model, for a direct comparison with our proposed models. In developing MixNAM, we constructed a multi-channel, fully connected network capable of encoding multiple features independently in a parallel manner.

For a fair comparison with existing methods, our experiments employ the processed version of MIMIC-II, MIIMC-II, Income, and Credit from (Chang et al., 2022), where the datasets are split into five parts for a five-fold cross-validation. The Housing dataset and the Year dataset are split into the training, validation, and test sets following setups of previous research (Chang et al., 2022;

---

[1]The Credit dataset can be downloaded at `https://www.kaggle.com/datasets/mlg-ulb/creditcardfraud`.

[2]The Year dataset can be downloaded at `https://archive.ics.uci.edu/dataset/203/yearpredictionmsd`

Radenovic et al., 2022), and each of them is tested with 10 different seeds to examine the variation of predictions. Consistent with previous studies (Chang et al., 2022; Popov et al., 2020; Radenovic et al., 2022), we applied the same quantile transformation to the features. Our evaluation metrics include the Root Mean Square Error (RMSE) for the Housing dataset, the Area Under the Receiver Operating Characteristic Curve (AUC) for the Income, MIMIC-II, and MIMIC-III datasets, and the Mean Square Error (MSE) for the Year dataset following existing research (Chang et al., 2022; Popov et al., 2020; Radenovic et al., 2022).

To be consistent with existing NAMs (Agarwal et al., 2021; Radenovic et al., 2022), the feature encoders $f_1, \cdots, f_n$ in MixNAM are implemented with MLPs. The expert predictors $E_{11}, \cdots, E_{nC}$ are implemented as one linear layer in our experiments as mentioned in Section 3.1. During training, we decrease the learning rate with cosine annealing following NBM (Radenovic et al., 2022). AdamW optimizer (Loshchilov & Hutter, 2018) is used to optimize the training objective.

All experiments were run on NVIDIA A100 GPUs (40 GB / 80 GB).

## C  HYPERPARAMETERS

We randomly search hyperparameters in the following ranges:

- #layers: the number of layers in each feature encoder, sampled from $\{3, 4\}$.
- Hidden dimension: the number of nodes in each layer of the feature encoder, sampled from $\{64, 128\}$.
- #total experts: the number of total experts for each feature, sampled from $\{4\}$.
- #activated experts: the number of activated experts for each feature, sampled from $\{4\}$.
- Batch size: the number of samples in one batch during training, sampled from $\{512, 1024, 2048\}$.
- Max iteration: the number of iterations for training, sampled from $\{75, 150, 500, 1000\}$.
- Learning rate: the speed of gradient descent, sampled from [1e-6, 1e-1].
- Weight decay: the coefficient for the L2 normalization on parameters, sampled from [1e-8, 1e-1].
- Dropout: the probability of a parameter in feature encoders being replaced as 0 during training, sampled from $\{0.0, 0.1, 0.2, 0.3, 0.4, 0.5, 0.6, 0.7, 0.8, 0.9\}$.
- Dropout expert: the probability of an expert's output being replaced as 0 during training, sampled from $\{0.1, 0.2, 0.3, 0.4, 0.5, 0.6, 0.7, 0.8, 0.9\}$.
- Output penalty: the coefficient for the L2 normalization on the outputs of features for reducing unimportant ones, sampled from [1e-8, 1e-1].
- Variation penalty: the weight for the expert variation penalty, sampled from $\{0.1\}$.
- Normalization: normalization methods used in the network, sampled from {batch_norm, layer_norm}.

The best hyperparameters we found for MixNAM are shown in Tables 5.

## D  MORE RELATED WORK ON GENERALIZED ADDITIVE MODELS WITH MIXTURE OF EXPERTS

In addition to the general related work on the design of generalized additive models (GAMs), plenty of efforts have been made in the application of GAMs (Hastie & Tibshirani, 1995; Sapra, 2013; Pedersen et al., 2019; Izadi, 2021), the evaluation of GAMs (Hegselmann et al., 2020; Chang et al., 2021), and the generalization of GAMs to high-order feature interaction modeling (Enouen & Liu, 2022; Tan et al., 2018). In the context of Mixture of Experts (MoE), there is some specific related work that tried to combine the idea of MoE with GAMs. Below we will introduce these attempts and discuss the differences between them and our MixNAM.

Table 5: Hyperparameters for MixNAM on all datasets.

| Hyperparameter | Housing | MIMIC-II | MIMIC-III | Income | Credit | Year |
|---|---|---|---|---|---|---|
| #layers | 4 | 4 | 4 | 4 | 4 | 4 |
| Hidden dimension | 128 | 128 | 128 | 128 | 128 | 128 |
| #total experts | 4 | 4 | 4 | 4 | 4 | 4 |
| #activated experts | 4 | 4 | 4 | 4 | 4 | 4 |
| Batch size | 2048 | 2048 | 1024 | 2048 | 2048 | 512 |
| Max iteration | 1000 | 1000 | 500 | 1000 | 150 | 75 |
| Learning rate | 5.97e-4 | 1.97e-4 | 1.68e-4 | 1.11e-4 | 4.00e-5 | 1.17e-4 |
| Weight decay | 5.29e-5 | 8.06e-4 | 2.78e-4 | 4.25e-3 | 1.88e-3 | 2.28e-6 |
| Dropout | 0.1 | 0.0 | 0.1 | 0.0 | 0.3 | 0.1 |
| Dropout expert | 0.2 | 0.4 | 0.2 | 0.5 | 0.2 | 0.6 |
| Output Penalty | 1.97e-5 | 3.49e-8 | 4.86e-5 | 3.99e-1 | 5.07e-4 | 1.45e-4 |
| Variation Penalty | 0.1 | 0.1 | 0.1 | 0.1 | 0.1 | 0.1 |
| Normalization | layer_norm | layer_norm | layer_norm | layer_norm | batch_norm | layer_norm |

GAMLSS (Rigby & Stasinopoulos, 2005) uses additive models to learn the distribution parameters of the target output such as mean and variance. While it is more flexible than standard GAMs, GAMLSS is still a strictly additive model. For example, consider a Gaussian output $y \sim \mathcal{N}(\mu, \sigma^2)$ modeled by $\mu = \sum_{i=1}^{n} f_i(x_i)$ and $\sigma = \sum_{i=1}^{n} g_i(x_i)$. With the reparameterization trick, the prediction can be re-written as

$$y = \mu + \sigma\varepsilon = \sum_{i=1}^{n} \Big( f_i(x_i) + \varepsilon g_i(x_i) \Big), \tag{15}$$

where $x_1, \cdots, x_n$ are the $n$ features and $\varepsilon \sim \mathcal{N}(0, 1)$. It can be seen that the output is still a strict addition of different feature information. Similarly, FlexMix (Leisch, 2004) provides a mixture of GAMs where the expert weights are learned and fixed by an EM algorithm. These two models are still strictly additive in general.

Mixdistreg (Rügamer, 2023; Rügamer et al., 2024) models flexible mixture weights dynamically determined by the input. Actually, it can be considered as a simplified version of our MixNAM. In Mixdistreg, the mixture of GAMs can be modeled as

$$F(x) = \sum_{k=1}^{K} \Big( \pi_k(x) \sum_{i=1}^{n} f_{ki}(x_i) \Big) = \sum_{i=1}^{n} \Big( \sum_{k=1}^{K} \pi_k(x) f_{ki}(x_i) \Big), \tag{16}$$

where $K$ is the number of experts and $\pi_1(x), \cdots, \pi_K(x)$ are the learned weights for each expert given $x$. While the weights are dynamically learned, they are kept the same across different features, which is less flexible than our MixNAM, where the learned weights could be diverse for different features. Also, the control of accuracy-interpretability trade-offs using our proposed variation penalty is not introduced in the previous work.

In general, classic methods such as GAMLSS (Rigby & Stasinopoulos, 2005) and FlexMix (Leisch, 2004) are strict additive models, which cannot mitigate the additivity constraint to achieve a balance between model performance and interpretability. Meanwhile, the recent Mixdistreg (Rügamer, 2023; Rügamer et al., 2024) mitigates the constraint but can be considered a simplified version of our MixNAM, with no control on the trade-offs between model accuracy and interoperability.

In addition to additive models, another prevalent approach for interpretability in tabular data involves post-hoc feature attribution. Methods like LIME (Ribeiro et al., 2016) and SHAP (Scott et al., 2017) offer local explanations by detailing feature contributions for individual predictions. However, MixNAM stands apart by providing a transparent, global understanding of how features influence predictions across the entire dataset.

## E  ADDITIONAL SIMULATION STUDIES ON EXTREME CASES

### E.1  SIMULATION OF FEATURES WITH HIGH SPARSITY

In addition to the simulation study presented in Section 4.4, we further explore how MixNAM performs when processing data with highly sparse features. Following the original simulation settings, the output of the simulated multimodal data is computed as:

$$y = x_2 \sin(4\pi x_1) + x_2 + \varepsilon \qquad \text{with} \qquad \varepsilon \sim \mathcal{N}(0, 0.1^2). \tag{17}$$

Different from the previous simulation where $x_2$ is sampled from {-1, 1} with equal probabilities, we manually control the proportion of data points with $x_2 = 1$ to represent 25%, 5%, and 1% of the dataset.

Figure 6 shows the learned shape plots of NAM and MixNAM on simulated data with different sparsities. We also explore how the number of experts $C$ will affect the shape function learning of MixNAM. For simplicity, all experts will be activated during the expert routing (i.e., $K = C$). The results demonstrate that NAM tends to focus on the majority group when the signal for $x_2 = 1$ is sparse. In contrast, MixNAM successfully identifies both modalities in the data distribution and learns the sparse distribution better with an increasing number of experts.

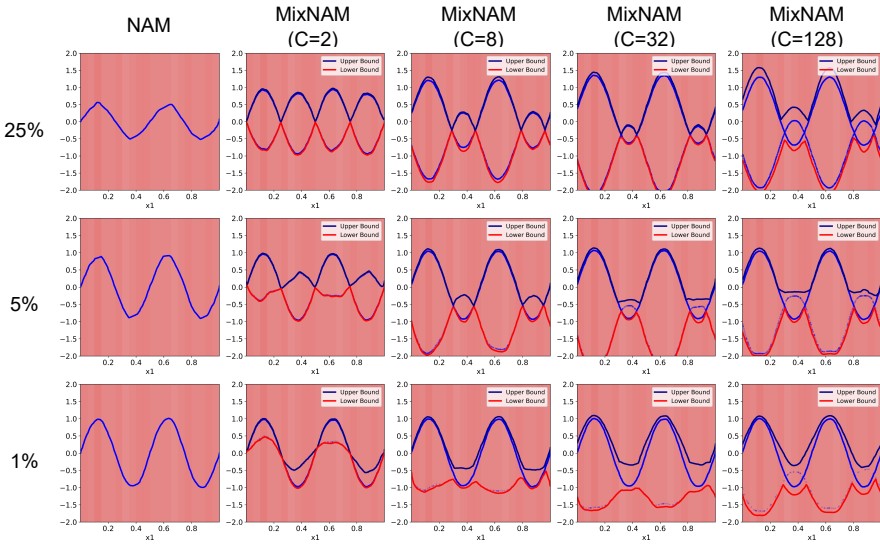

Figure 6: Shape plots of MixNAM and NAM on the simulated data with high sparsity.

### E.2  SIMULATION OF FEATURES WITH HIGH VARIABILITY

As we introduced above, the variability in model prediction refers to the extent to which the predicted outputs of a model fluctuate or differ when the same input feature value is presented across different samples. Formally, consider the mapping from the input features $x_1, \cdots, x_n$ to the predicted output:

$$\hat{y} = F(x_1, x_2, \cdots, x_n), \tag{18}$$

where $F$ represents the underlying predictive function. For a fixed value of $x_i = a$, the variability of the contribution of $x_i$ to $\hat{y}$ can be defined as:

$$\text{variability}|_{x_i=a} = \text{Var}_{x_1, \cdots, x_n}[F(x_1, x_2, \cdots, x_n | x_i = a) - \text{E}_{x_i}(F(x_1, x_2, \cdots, x_n))]. \tag{19}$$

This definition measures how the contributions of $x_i = a$ deviate due to interactions with other features, encapsulating variability caused by feature dependencies. For traditional additive models, the variability defined above reduces to zero because the contributions of each feature are independent

and do not interact. In models where interactions exist, this term captures the extent to which other features $x_j (j \neq i)$ influence the contribution of $x_i$.

To assess the performance of MixNAM under conditions of extreme variability in output contributions caused by a large number of modalities, we simulate a data distribution using the following equation:

$$y = \varepsilon + \sum_{i=2}^{NC+1} \frac{T}{NC} x_i \sin(4\pi x_1), \tag{20}$$

where the feature $x_1$ is sampled from $U(0,1)$ and $x_2, \cdots, x_{NC+1}$ are the $NC$ categorical features sampled from $\{-1, +1\}$ with equal probability. $T$ is a scaling factor that amplifies variability across different modalities. For this study, we set $T = 64$ and evaluated both NAM and MixNAM across scenarios with $NC = 1, 2, 4, 8, 16$. The number of simulated samples is set dynamically as $2000 \times NC$.

As illustrated in Figure 7, the results demonstrate that MixNAM effectively captures the multimodal contributions of $x_1$ to the output, showing consistent performance as the number of features and modalities increases. In contrast, NAM struggles to account for multimodality due to its inherent limitation of feature additivity. The results highlight the robustness and scalability of MixNAM in handling multimodal data with high variability.

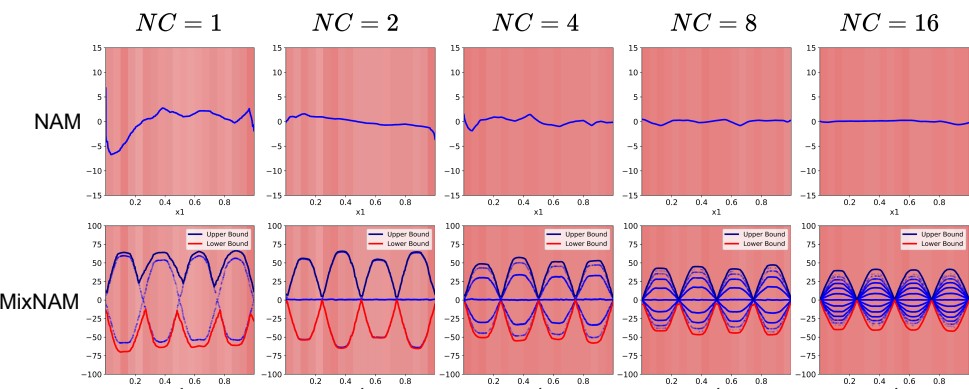

Figure 7: Shape plots of MixNAM and NAM on the simulated data with high variability.

## F ADDITIONAL ANALYSIS OF EXPERT ROUTING IN MIXNAM

While the expert routing mechanism in MixNAM can capture certain feature interactions and provide a continuous output range between the lower and upper bounds for each feature, it should be noted that the router is not a universal function approximator that dominates the model prediction even if there is no variation penalty.

Specifically, if each feature has only two expert functions to model the upper and lower bounds, both of which are activated, the relevant score estimation of each expert is a normalized Generalized Additive Model (GAM) with the sigmoid function as the link function. To prove this, we can rewrite the relevance score for the first expert of the $i$-th feature (Formula 6) as

$$
\begin{aligned}
r_{i1} &= \frac{\exp(\varphi_i[1])}{\exp(\varphi_i[1]) + \exp(\varphi_i[2])} \\
&= \frac{\exp(e_1^\top \varphi_i)}{\exp(e_1^\top \varphi_i) + \exp(e_2^\top \varphi_i)} \qquad\qquad (e_1 := [1,0]^\top, e_2 := [0,1]^\top) \\
&= \frac{\exp((e_1 - e_2)^\top \varphi_i)}{\exp((e_1 - e_2)^\top \varphi_i) + 1} \\
&= \sigma((e_1 - e_2)^\top \varphi_i) \qquad\qquad (\sigma(\cdot) := \mathrm{Sigmoid}(\cdot)) \\
&= \sigma\Big((e_1 - e_2)^\top (\mu_i + \sum_{j=1}^n A_{ji}^\top f_j(x_j))\Big) \qquad\qquad (\text{Formula 4}) \\
&= \sigma\Big(e_1^\top \mu_i - e_2^\top \mu_i + \sum_{j=1}^n \Big(e_1^\top A_{ji}^\top f_j(x_j) - e_2^\top A_{ji}^\top f_j(x_j)\Big)\Big).
\end{aligned}
\tag{21}
$$

With

$$
g_{i0} := e_1^\top \mu_i - e_2^\top \mu_i \quad \text{and} \quad g_{ij}(\cdot) := e_1^\top A_{ji}^\top f_j(\cdot) - e_2^\top A_{ji}^\top f_j(\cdot),
$$

the relevance score can be further rewritten as

$$
r_{i1} = \sigma\Big(g_{i0} + \sum_{j=1}^n g_{ij}(x_j)\Big),
$$

which is a normalized GAM where the linked function is a sigmoid function. Thus, instead of interpolating all values between the upper and lower bounds, the expert routing mechanism can only interpolate certain positions whose values are restricted by a normalized GAM.

Similarly, for more general cases with $K$ activated experts among $C$ experts, the estimated relevance for the activated experts can be considered $K$ probability values normalized by the softmax function. By definition in Formula 4, each probability before the normalization is given by a GAM.

Such an analysis can be verified by the result of our additivity evaluation in Section 4.3.2, where the additivity of MixNAM is significantly higher than MLP, even when there is no variation penalty.

## G  MixNAM-D: MixNAM with Diagonal Routing Matrix

As mentioned in Section 3.2, we explored a special version of MixNAM, termed MixNAM-D, where the scoring matrics compose a block diagonal matrix. In MixNAM-D, the matrix $\mathcal{A}_{ij}$ in Formula 4 becomes a zero matrix if $i \neq j$. This configuration allows MixNAM-D to function similarly to NAMs, where the relevance estimation of a feature's experts is solely based on its own encoded information.

Since the relevance for experts of each feature is solely determined by itself in MixNAM-D, the estimated relevance scores will be fixed for the same feature value. To encourage the model to learn various distributions with multiple experts, we introduce randomness into the decision-making process by employing the Gumbel-softmax technique (Jang et al., 2016) with the temperature $\tau = 0.1$ to re-sample the experts during training based on their estimated relevance scores in Formula 6:

$$
\hat{r}_{ik} = \exp\left(\frac{\log(r_{ik}) + g_i}{\tau}\right) \Big/ \sum_{l=1}^C \exp\left(\frac{\log(r_{il}) + g_l}{\tau}\right), \qquad \text{where } g_i, g_l \sim Gumbel(0,1). \tag{22}
$$

The performance comparison of MixNAM-D with other baselines is presented in Table 6. The results demonstrate that, by regularizing the scoring matrix to be diagonal, MixNAM-D shows a performance close to traditional additive models, which have worse accuracies than complex models that capture feature interactions.

Figure 8 shows the shape plots generated by MixNAM-D compared to other baseline additive models. As discussed above, MixNAM-D is a strictly additive model where the effect of each feature on

Table 6: Comparison of MixNAM-D to other baselines on benchmark datasets. "FA" (Feature Attribution) capabilities of different models are denoted by ✔for presence and ✘ for absence.

| Model | FA | Housing | MIMIC-II | MIMIC-III | Income | Credit | Year |
|---|---|---|---|---|---|---|---|
| | | RMSE ↓ | AUC ↑ | AUC ↑ | AUC ↑ | AUC ↑ | MSE ↓ |
| MLP | ✘ | 0.501 ± 0.006 | 0.835 ± 0.014 | 0.815 ± 0.009 | 0.914 ± 0.003 | 0.981 ± 0.007 | 78.48 ± 0.56 |
| XGBoost | ✘ | 0.443 ± 0.000 | 0.844 ± 0.012 | 0.819 ± 0.004 | 0.928 ± 0.003 | 0.978 ± 0.009 | 78.53 ± 0.09 |
| EB$^2$M | ✘ | 0.492 ± 0.000 | 0.848 ± 0.012 | 0.821 ± 0.004 | 0.928 ± 0.003 | 0.982 ± 0.006 | 83.16 ± 0.01 |
| NA$^2$M | ✘ | 0.492 ± 0.008 | 0.843 ± 0.012 | 0.825 ± 0.006 | 0.912 ± 0.003 | 0.985 ± 0.007 | 79.80 ± 0.05 |
| EBM | ✔ | 0.559 ± 0.000 | 0.835 ± 0.011 | 0.809 ± 0.004 | 0.927 ± 0.003 | 0.974 ± 0.009 | 85.81 ± 0.11 |
| NAM | ✔ | 0.572 ± 0.005 | 0.834 ± 0.013 | 0.813 ± 0.003 | 0.910 ± 0.003 | 0.977 ± 0.015 | 85.25 ± 0.01 |
| MixNAM-D | ✔ | 0.553 ± 0.001 | 0.830 ± 0.012 | 0.805 ± 0.006 | 0.927 ± 0.003 | 0.977 ± 0.010 | 85.60 ± 0.04 |

the final output is determined by the feature itself. Thus, we plot MixNAM-D in the same way as we did for EBM and NAM, directly showing the estimated contribution given by each feature value. Moreover, since MixNAM-D is trained under the general MixNAM framework with multiple experts learned for each feature, we plot another figure for MixNAM-D including the estimated upper and lower bounds based on the learned experts.

It can be observed from Figure 8 that the patterns captured by MixNAM-D are similar to the ones by EBM and NAM. Beyond the point estimation provided by traditional additive models, MixNAM-D offers additional insights with its prediction bounds. In areas with low data density, the plot displays a wide gap between the upper and lower bounds, highlighting variability in predictions where less data is available.

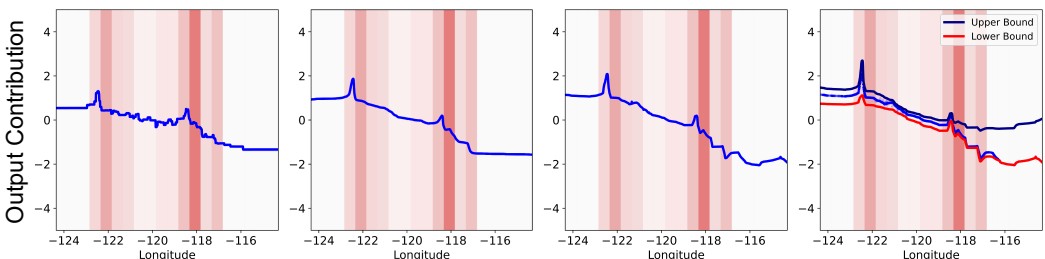

Figure 8: Visual comparison of the "Longitude" feature impact on house prices by MixNAM-D.

# H MIXNAM-E: MIXNAM WITH EVENLY DISTRIBUTED EXPERT ACTIVATION

While the activation strategy detailed in Section 3.2 facilitates continuous expert relevance estimation, we propose an adaptation that shifts this estimation to a discrete framework. This modification involves adjusting the weights to be evenly distributed across relevant experts, leading to a modified model termed MixNAM-E. To implement this, we use the masking vector outlined in Formula 5 and modify the relevance computation in Formula 6 as follows:

$$r_{ik} = \exp(\varphi_i[k] - \varphi_i^*[k] + M_i[k]) \Big/ \sum_{l=1}^{C} \exp(\varphi_i[l] - \varphi_i^*[l] + M_i[l]). \tag{23}$$

$\varphi_i^*$ mirrors the values of $\varphi_i$ but does not require gradient computations. The activation strategy implemented by Formula 23 will result in a finite set of possible outcomes for each input feature value. Given a configuration of $C$ experts and $K$ activated ones per feature, the predicted output by MixNAM-E for any given feature will be one of the possible selections from $\binom{C}{K}$ combinations.

Table 7 shows the performance (RMSE) of MixNAM-E on the Housing dataset with different variation penalties, along with their corresponding shape plots of the "Longitude" feature. All the tested MixNAM-E models have a configuration of $C = 32$ and $K = 16$. In the plots, we illustrate the upper and lower bounds for each feature by selecting the maximum and minimum values from the $\binom{C}{K}$ possible combinations.

Compared with MixNAM (Table 2), the performance of MixNAM-E is slightly worse due to the limitations imposed by its discrete range. However, even without any variation penalty, the estimated bounds by MixNAM-E still fit the actual score distribution tightly, accurately reflecting the prediction limits.

Table 7: Performance (RMSE↓) and explanations of MixNAM-E with different variation penalties.

| $\lambda = 0$ | $\lambda = 0.1$ | $\lambda = 1$ | $\lambda = 10$ | $\lambda = 100$ |
|---|---|---|---|---|
| RMSE ($\downarrow$) | | | | |
| $0.462 \pm 0.002$ | $0.460 \pm 0.002$ | $0.483 \pm 0.001$ | $0.551 \pm 0.002$ | $0.585 \pm 0.003$ |
| Shape Plot | | | | |

# I  PERFORMANCE OF MIXNAM WITH THE SCALING OF EXPERTS

As MixNAM employs a mixture of experts for data modeling, we explore how the number of total experts ($C$) and the number of activated experts ($K$) affect the overall model performance. Table 8 presents the performance of both MixNAM and MixNAM-E on the Housing dataset under various configurations. The results indicate that MixNAM reaches optimal performance at $C = 8$ and $K = 4$, and further increases in $C$ or $K$ do not enhance its performance. Such a finding is consistent with the role of experts in MixNAM, which primarily explore the variability in the prediction space to learn the upper and lower bounds, as the dynamic routing mechanism offers a continuous range within the bounds for prediction. In contrast, MixNAM-E benefits from an increased number of $C$ and $K$, showing improved performance as these parameters grow. This difference shows how the continuous expert activation in MixNAM contrasts with the discrete one in MixNAM-E, highlighting the flexibility of our MixNAM framework to adapt to various scenarios through different implementations.

Table 8: Results for different numbers of activated experts ($K$) and all experts ($C$).

| $K \setminus C$ | MixNAM | | | | MixNAM-E | | | |
|---|---|---|---|---|---|---|---|---|
| | 2 | 4 | 8 | 16 | 16 | 32 | 64 | 128 |
| 2 | 0.479 $\pm$ 0.003 | 0.459 $\pm$ 0.003 | 0.455 $\pm$ 0.004 | 0.460 $\pm$ 0.004 | 0.486 $\pm$ 0.007 | 0.487 $\pm$ 0.005 | 0.486 $\pm$ 0.006 | 0.489 $\pm$ 0.005 |
| 4 | – | 0.451 $\pm$ 0.002 | 0.447 $\pm$ 0.004 | 0.455 $\pm$ 0.004 | 0.469 $\pm$ 0.007 | 0.466 $\pm$ 0.005 | 0.468 $\pm$ 0.004 | 0.470 $\pm$ 0.004 |
| 8 | – | – | 0.449 $\pm$ 0.004 | 0.456 $\pm$ 0.003 | 0.470 $\pm$ 0.003 | 0.459 $\pm$ 0.004 | 0.458 $\pm$ 0.002 | 0.458 $\pm$ 0.004 |
| 16 | – | – | – | 0.456 $\pm$ 0.005 | 0.587 $\pm$ 0.003 | 0.462 $\pm$ 0.002 | 0.451 $\pm$ 0.002 | 0.453 $\pm$ 0.002 |

## J    DISCUSSION ON MODEL COMPLEXITY

As MixNAM is a fundamental framework aiming to extend traditional additive models, our design prioritizes the performance and comprehensiveness of the overall framework over the optimization of model efficiency. On the basis of Neural Additive Models (NAMs), MixNAM introduces more parameters in the expert encoding and dynamic routing steps. To rigorously analyze how much cost MixNAM will bring compared to NAM, suppose we have $n$ features, $d$-dimensional output for feature encoding, and $C$ total experts for each feature. In our experiments, MixNAM is implemented with $d = 128$ and $C = 4$. For MixNAM-D we have $d = 128$ with $C = 64$. The theoretical and empirical additional costs brought by MixNAM and its variant are presented in Table 9, including the increase in parameter count and the additional memory required (assuming float32 storage).

Table 9: Theoretical and empirical additional costs brought by MixNAM compared to NAM. Mix-NAM is implemented with $d = 128, C = 4$. MixNAM-D is implemented with $d = 128, C = 64$.

|  | #Features | MixNAM | | MixNAM-D | |
|---|---|---|---|---|---|
|  |  | **Parameter Count** | **Memory Usage** | **Parameter Count** | **Memory Usage** |
| Housing | 8 | 37k | 144k | 132k | 516k |
| MIMIC-II | 17 | 157k | 613k | 281k | 1.1M |
| MIMIC-III | 57 | 1.7M | 6.5M | 941k | 3.59M |
| Income | 14 | 108k | 420k | 231k | 903k |
| Credit | 30 | 476k | 1.8M | 495k | 1.89M |
| Year | 90 | 4.2M | 16.0M | 1.5M | 5.67M |
| Theoretical | – | $nC[(n+1)d+2]$ | | $nC(2d+2)$ | |

The results indicate that the additional cost associated with MixNAM-D scales linearly with the number of features, whereas the cost for MixNAM includes a squared term due to its more complex routing mechanism. Specifically, MixNAM utilizes a full $n \times n$ block matrix to encode feature interactions, which is simplified as a block diagonal matrix in MixNAM-D. Despite this increased complexity, the additional memory usage remains manageable with current computing resources. Future efforts could be made to sparsify the routing matrix in MixNAM, which could potentially reduce the additional cost while retaining the model performance.

## K    MORE VISUALIZATION RESULTS OF MIXNAM ON REAL-WORLD DATA

Figures 9 - 12 show the complete visualization results of feature explanations by MixNAM on different datasets. Here we present results for datasets with no more than 50 features, including Housing, MIMIC-II, Income, and Credit.

## L    DISCUSSIONS, LIMITATIONS AND FUTURE WORK

MixNAM can have a significant impact on high-stakes areas such as finance and healthcare where the explanation of predictions is as crucial as the predictions themselves. The introduction of Mix-NAM advances the capabilities of additive models, which may encourage the deployment of related research in practical applications. Moreover, MixNAM generalizes existing Neural Additive Models, offering a new direction to enhance additive models by balancing the accuracy and interpretability under the MixNAM framework. However, MixNAM does not inherently address issues of fairness or mitigate implicit bias in the data. Instead, it provides transparent feature explanations that may highlight these biases. Therefore, careful considerations and responsible use are essential when applying MixNAM in scenarios where fairness is a critical concern. The transparency of Mix-NAM may also serve as a tool for identifying data biases, contributing to equitable and accountable AI systems.

We acknowledge that this work has certain limitations, which could be the direction of future research. First, while MixNAM provides a comprehensive visualization of feature impacts on predictions, the intervenability of additive models is reduced due to the uncertainty in outcome predictions

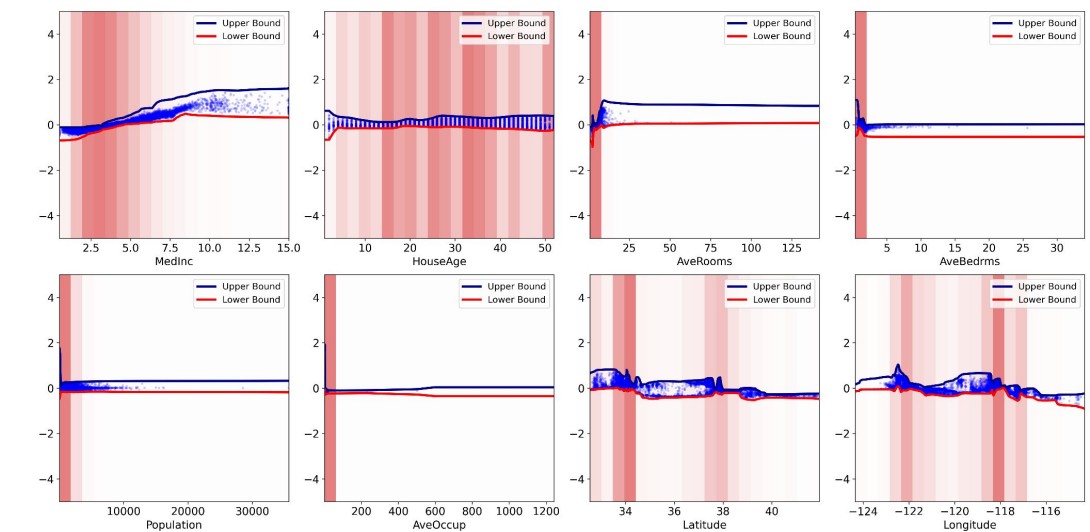

Figure 9: Visualization of feature explanations by MixNAM on the Housing dataset.

when altering single feature values. However, MixNAM compensates by providing a range of possible outcomes, defined by estimated upper and lower bounds. The expert variation penalty can also adjust trade-offs between intervenability and performance. Future efforts could be made to handle the uncertainty in MixNAM to offer precise estimations after interventions. Second, although recent advancements have improved the efficacy and scalability of Neural Additive Models (Radenovic et al., 2022; Zhang et al., 2024), our MixNAM is constructed based on the foundational vanilla NAM, which serves as the most general framework for neural-network-based additive models. Enhancing the efficiency of MixNAM could be a valuable direction for subsequent research. Third, we have tested our MixNAM only on tabular data, since features are clearly defined in this modality, aligning with prior studies on additive models (Agarwal et al., 2021; Chang et al., 2022; Lou et al., 2012). However, our framework has the potential to be generalized to other modalities. For example, features in images can be extracted with object detection (Bochkovskiy et al., 2020; Tan et al., 2020) or concept learning methods (Ghorbani et al., 2019), and the feature encoders in MixNAM could be implemented with CNN (He et al., 2016; Krizhevsky et al., 2012) or Transformer (Dosovitskiy et al., 2020; Vaswani et al., 2017) to process data in different modalities. We leave this as one potential direction for future research and applications.

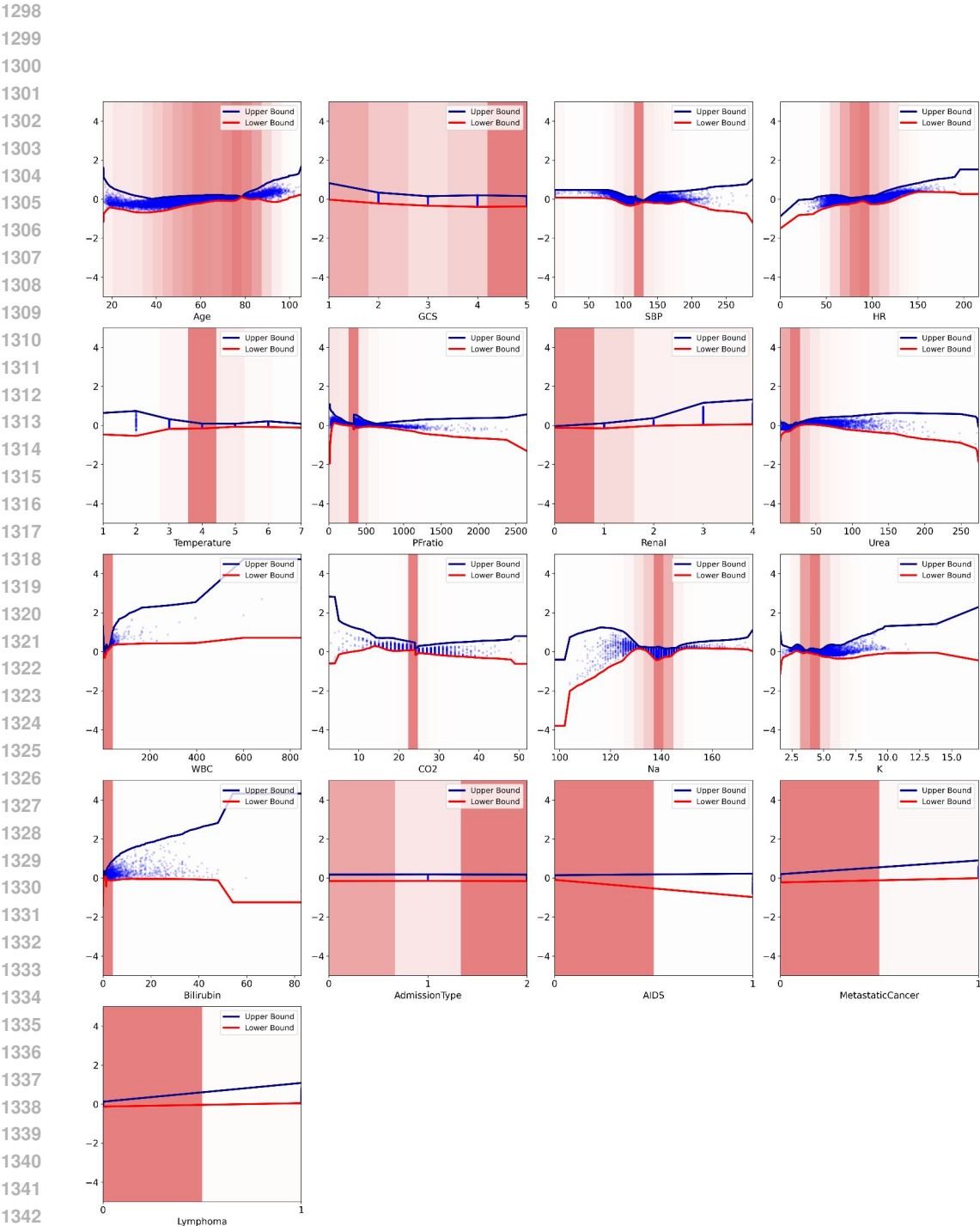

Figure 10: Visualization of feature explanations by MixNAM on the MIMIC-II dataset.

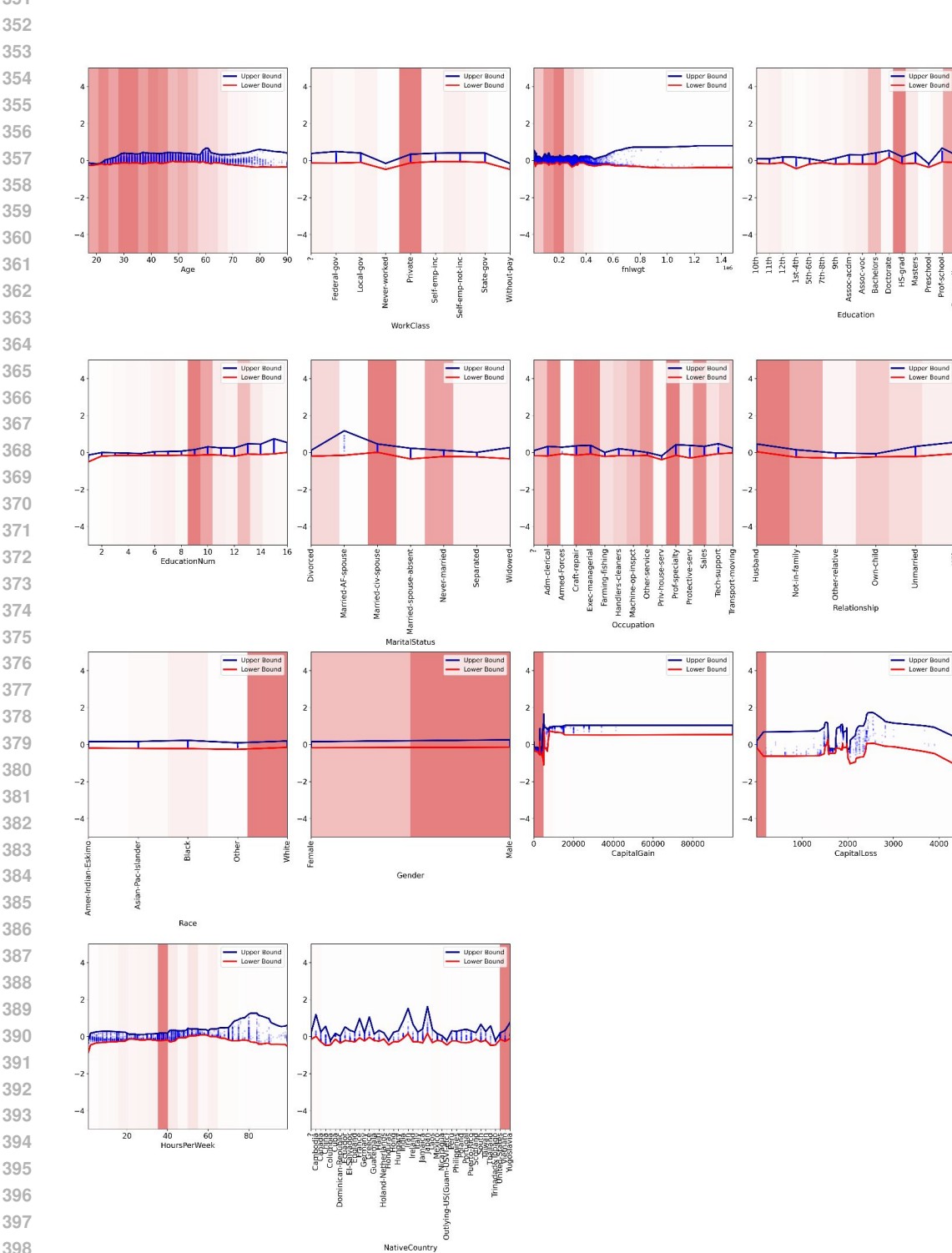

Figure 11: Visualization of feature explanations by MixNAM on the Income dataset.

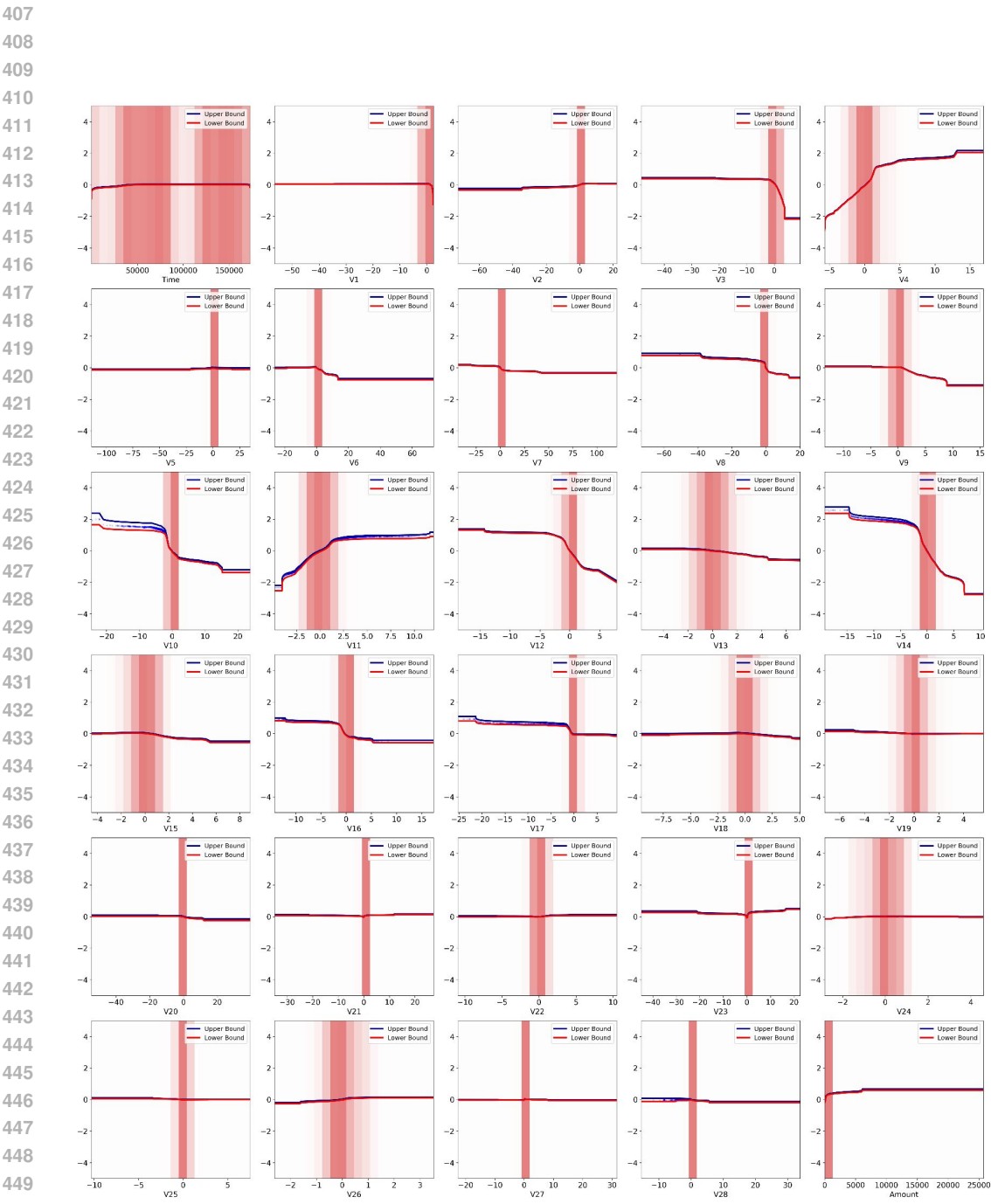

Figure 12: Visualization of feature explanations by MixNAM on the Credit dataset.

