# OpenReview forum: "MixNAM: Advancing Neural Additive Models with Mixture of Experts"
_ICLR.cc/2025/Conference — Submitted to ICLR 2025_

### Official Review · Reviewer_xFNJ · 2024-10-30

**Soundness:** 2
**Presentation:** 3
**Contribution:** 1
**Rating:** 3
**Confidence:** 4

**Summary:**

This paper introduces an enhancement to Neural Additive Models (NAMs) by incorporating a Mixture of Experts (MoE) framework, enabling the model to capture feature variability and complex feature interactions. This approach allows each feature to be modeled through multiple expert predictions, which are dynamically selected based on relevance, thus addressing traditional NAM limitations in handling real-world data complexity. This new framework enhances additive models' utility, offering advanced predictive accuracy and transparency.

**Strengths:**

1. The paper combines NAMs’ interpretability with MoE’s capacity for multi-aspect feature representation, contributing a new approach to interpretable machine learning.

2. MixNAM’s ability to capture variability in feature impacts is highly relevant for real-world applications that require transparent models with high predictive power. The proposed method also allows users to balance accuracy with interpretability.

**Weaknesses:**

1. As described in line 164, each expert $E_{ik}$ is implemented as a linear layer, but normally, MLPs are used as base models for experts, the authors should justify why choosing linear model as experts.

2. The novelty of this paper is quite limited, as they just apply regular MoE architecture on top of each feature's embedding and weighted combine them to obtain the outputs, even so, the authors did not provide a detailed rationale for choosing such a method.

3. In line 336, it seems that the uncertainty intervals are formulated by a group of expert predictions, this is not a natural way of producing prediction intervals, as the uncertainty should come from the model parameters or the data itself.

4. The multimodal experiments are all based on small-scale simulated datasets, the authors should benchmark its method on larger-scale multimodal benchmarks.

**Questions:**

How does the proposed method handle load imbalance when training MoE models? Is the expert variation penalty similar to the load imbalance loss function?

---

> ### Author Response · Authors · 2024-11-20
> **Response to Reviewer xFNJ (Part 1)**
>
> Thank you for your thoughtful feedback. Your expertise in Mixture of Experts (MoE) is evident and greatly appreciated. However, we feel that our core contribution—advancing additive models by integrating MoE to balance interpretability and performance—may not be recognized. Below, we provide detailed responses to your concerns and clarify the unique motivations and contributions of MixNAM.
>
> ### Weaknesses:
>
> > As described in line 164, each expert $E_{ik}$ is implemented as a linear layer, but normally, MLPs are used as base models for experts, the authors should justify why choosing linear model as experts.
>
> We respectfully disagree that the use of linear layers should be considered a weakness. Instead, it is a critical design choice to improve the efficiency of the architecture. While it would be natural to implement $C$ separate MLPs for the $C$ experts of each feature, this approach would scale the number of parameters by a factor of $C$. To address this, we share parameters among experts by implementing a single MLP followed by $C$ linear layers, as shown in Figure 1. This design significantly reduces the parameter overhead while preserving flexibility in the expert outputs. The experimental results in Table 1 validate the effectiveness of this design, demonstrating improved performance compared to traditional additive models.
>
> > The novelty of this paper is quite limited, as they just apply regular MoE architecture on top of each feature's embedding and weighted combine them to obtain the outputs, even so, the authors did not provide a detailed rationale for choosing such a method.
>
> **The focus of this research is not to apply the MoE framework arbitrarily but to advance the capabilities of additive models**. The primary motivation for this work stems from the limitations of existing additive models, which typically produce less accurate predictions than black-box models. As shown in Table 1, additive models (marked with "FA") consistently underperform compared to non-additive models (e.g., MLP, XGBoost).
>
> Our objective is to improve the performance of additive models while preserving their interpretability. By introducing a mixture of experts, we relax the strict additivity constraint by allowing feature interactions in the expert routing mechanism, while maintaining an additive structure in the overall prediction. Empirical results (Table 1) show that this approach significantly enhances performance, achieving results comparable to complex black-box models. At the same time, MixNAM retains interpretability by enabling visualizations of feature contributions (Figures 3, 9-12). This makes MixNAM a valuable alternative to unexplainable methods like XGBoost.
>
> > In line 336, it seems that the uncertainty intervals are formulated by a group of expert predictions, this is not a natural way of producing prediction intervals, as the uncertainty should come from the model parameters or the data itself.
>
> The uncertainty/variability addressed in this study does not refer to randomness in model predictions. Instead, we focus on the variability in how a feature's contribution, $x_i = a$, is influenced by other features, $x_j$ for $j \neq i$. Formally, this variability is defined as:
> $$variability_{x_i=a} = Var_{x_1,\cdots,x_n}[F(x_1, x_2, \cdots, x_n|x_i=a) - E_{x_i}(F(x_1, x_2, \cdots, x_n))].$$
> This term is zero in additive models, where features do not interact, but nonzero in scenarios with feature interactions. By modeling such variability with MoE, we aim to bridge the gap between interpretable but less accurate additive models and powerful but opaque black-box models.

---

> ### Author Response · Authors · 2024-11-20
> **Response to Reviewer xFNJ (Part 2)**
>
> > The multimodal experiments are all based on small-scale simulated datasets, the authors should benchmark its method on larger-scale multimodal benchmarks.
>
> Our current simulation analysis is designed as a qualitative demonstration of how MixNAM captures multimodal distributions that traditional additive models fail to model. Traditional additive models inherently produce deterministic outputs for each feature value, which limits their ability to handle multimodal data.
>
> For larger-scale multimodal benchmarks, we have added an additional simulation study in Section E.2 of our updated manuscript. This study evaluates MixNAM’s performance as the number of features, samples, and modalities increases. Specifically, we simulated a data distribution using the following pattern:
> $$y=\varepsilon + \sum_{i=2}^{NC+1}\frac{T}{NC}x_i\sin(4\pi x_1),$$
> where the feature $x_1$ is sampled from $U(0, 1)$ and $x_2,\cdots,x_{NC+1}$ are the $NC$ categorical features sampled from \{-1, +1\} with equal probability. $T$ is a scaling factor that amplifies variability across different modalities. For this study, we set $T = 64$ and evaluated both NAM and MixNAM across scenarios with $NC = 1, 2, 4, 8, 16$. The number of simulated samples is set dynamically as $2000\times NC$.
>
> The results, as presented in Figure 7 of the updated manuscript, demonstrate that MixNAM effectively captures the multimodal contributions of $x_1$ to the output, showing consistent performance as the number of features and modalities increases. It highlights the robustness and scalability of MixNAM in handling multimodal data with high variability.
>
> ### Questions:
> > How does the proposed method handle load imbalance when training MoE models? Is the expert variation penalty similar to the load imbalance loss function?
>
> We did not explicitly use a load imbalance loss function to address this issue. Instead, we included expert dropout, as mentioned in line 228, to prevent the model from over-relying on specific experts.

---

### Official Review · Reviewer_MzpU · 2024-10-31

**Soundness:** 3
**Presentation:** 3
**Contribution:** 3
**Rating:** 6
**Confidence:** 3

**Summary:**

The paper introduces MixNAM which improves neural additive model (NAMs) by combining with a mixture of experts (MoE) to capture the feature variability due to its interaction with other features. MixNAM uses multiple experts for each feature and uses a dynamic routing mechanism to assess and combine the relevance of different experts. This improves the prediction performance and gives the upper and lower bound of the feature attribution. The empirical evaluation demonstrates that MixNAM outperforms traditional additive models that ignore interactions and achieves comparable performance to complex black-box models while providing feature attributions.

**Strengths:**

1. The idea of combining MoE with NAM is novel and can improve the flexibility of NAM to capture feature interactions and maintain interpretability. The claim is supported by extensive experiments and benchmarks against various baselines, in terms of prediction performance and interpretability.

2. MixNAM provides a point estimator of feature attribution but a range of it, which reflects the feature interaction.

**Weaknesses:**

1. If we look at Table 1, MixNAM is only significantly better than other interpretable methods (with FA) on the Housing and Year dataset by taking the standard error into consideration.

2. The model is benchmarked only on tabular datasets due to the limited flexibility of NAM on structured datasets, even though it's still beneficial to discuss its potential usage on structured datasets, as they are the primary use cases of neural networks.

3. The performance and interoperability of MixNAM are sensitive to the penalty parameter \lambda. The paper would benefit from a more in-depth discussion on how to choose it.

**Questions:**

1. How was the hyper-parameter \lambda selected? Was it cross-validated against prediction accuracy? It would be great to provide some heuristics to select it to balance the accuracy and interpretability.

2. It would be great to compare the interpretation of MixNAM with any post-hoc feature attribution (e.g., SHAP) on XGBoost, which is a popular choice to gain interpretability on tabular data. What is the difference between these two options? A practitioner may like to know when they should use an interpretable MixNAM and when they should use XGBoost with post-hoc interpretation methods.

**Details Of Ethics Concerns:**

No concerns of ethics.

---

> ### Author Response · Authors · 2024-11-20
> **Response to Reviewer MzpU (Part 1)**
>
> We sincerely appreciate your valuable feedback and thoughtful suggestions. Here are our detailed responses to your questions and conerns.
>
> ### Weaknesses:
> > If we look at Table 1, MixNAM is only significantly better than other interpretable methods (with FA) on the Housing and Year dataset by taking the standard error into consideration.
>
> This discrepancy arises because we performed experiments with 10 different seeds for the Housing and Year datasets, while for other datasets, we followed previous research [1,2] and used five-fold cross-validation. Cross-validation introduces larger standard deviations due to differences in data splits. Although statistical significance may appear less clear due to these variations, Table 1 demonstrates that MixNAM performs on par with complex models without feature attribution (FA), which are more expressive and powerful than traditional additive models.
>
> [1] NODE-GAM: neural generalized additive model for interpretable deep learning. ICLR 2022.
>
> [2] Neural basis models for interpretability. NeurIPS 2022.
>
> > The model is benchmarked only on tabular datasets due to the limited flexibility of NAM on structured datasets, even though it's still beneficial to discuss its potential usage on structured datasets, as they are the primary use cases of neural networks.
>
> We assume you meant "unstructured data" rather than "structured data." Please correct us if this interpretation is wrong.
>
> In the existing literature on additive models, evaluations primarily focus on tabular data, while applications to other modalities, such as text or images, are rare [1,2,3]. This is because the core capability of additive models lies in interpreting features through shape plots, which illustrate how predictions change as feature values monotonically increase or decrease. Additive models are difficult to evaluate on raw text or image data, where features and their monotonic relationships are challenging to define.
>
> For example, NBM [2] tested its performance on image data by using concept bottleneck models to preprocess images into tabular data. However, the effectiveness of the overall system was limited by the quality of the extracted concepts, which may not fully capture the information in the original images [4,5]. Therefore, we follow the mainstream in additive model research and evaluate MixNAM using tabular data. We have discussed the potential generalization of MixNAM to other modalities as a future direction in Appendix L (Appendix K).
>
> [1] Agarwal R, et al. Neural additive models: Interpretable machine learning with neural nets. NeurIPS 2021.
>
> [2] Radenovic F, et al. Neural basis models for interpretability. NeurIPS 2022.
>
> [3] Chang CH, et al. NODE-GAM: Neural Generalized Additive Model for Interpretable Deep Learning. ICLR 2022.
>
> [4] Koh PW, et al. Concept bottleneck models. ICML 2020.
>
> [5] Margeloiu A, et al. Do concept bottleneck models learn as intended?. 2021.
>
> > The performance and interoperability of MixNAM are sensitive to the penalty parameter \lambda. The paper would benefit from a more in-depth discussion on how to choose it.
>
> Table 2 and Table 3 present our further analysis of the $\lambda$ selection from both qualitative and quantitative perspectives. They illustrate how $\lambda$ effectively balances interpretability and performance in MixNAM. Based on our analysis of $\lambda$ on the Housing dataset (Section 4.3), we selected $\lambda=0.1$ as the default value in our main experiments, which turns out to perform robustly on all datasets with improved accuracy (Table 1) and retained interpretability (Figures 9-12).

---

> ### Author Response · Authors · 2024-11-20
> **Response to Reviewer MzpU (Part 2)**
>
> ### Questions:
>
> > How was the hyper-parameter \lambda selected? Was it cross-validated against prediction accuracy? It would be great to provide some heuristics to select it to balance the accuracy and interpretability.
>
> We analyzed different $\lambda$ values on the Housing dataset with respect to accuracy, additivity, and tightness (Table 3), as well as visual outputs (Table 2). These results showed that $\lambda = 0.1$ represents a "sweet spot," enabling MixNAM to significantly outperform strictly additive models while preserving interpretability through faithful shape plots with tight estimated bounds. Experiments on other datasets further validated the robustness of $\lambda = 0.1$, which consistently achieved improved accuracy (Table 1) and retained interpretability (Figures 9-12).
>
> > It would be great to compare the interpretation of MixNAM with any post-hoc feature attribution (e.g., SHAP) on XGBoost, which is a popular choice to gain interpretability on tabular data. What is the difference between these two options? A practitioner may like to know when they should use an interpretable MixNAM and when they should use XGBoost with post-hoc interpretation methods.
>
> Thank you for this great suggestion! Post-hoc feature attribution methods, such as LIME [1] and SHAP [2], provide local explanations, describing the contributions of features for individual predictions. In contrast, MixNAM provides a transparent and global understanding of how features influence predictions across the dataset. MixNAM advances traditional additive models by overcoming performance constraints while maintaining intrinsic interpretability.
>
> In practice, MixNAM is ideal when both global feature explanations and high model accuracy are required, while XGBoost with post-hoc methods may suffice if localized interpretability alone is adequate.
>
> [1] "Why Should I Trust You?": Explaining the Predictions of Any Classifier. KDD 2016.
>
> [2] A Unified Approach to Interpreting Model Predictions. NeurIPS 2017.

---

> ### Comment · Reviewer_MzpU · 2024-12-03
>
> Thank you authors for the detailed response and experiments. Most of my questions have been mostly solved so I keep my score positive.

---

### Official Review · Reviewer_NxyM · 2024-11-04

**Soundness:** 3
**Presentation:** 2
**Contribution:** 3
**Rating:** 6
**Confidence:** 3

**Summary:**

The paper introduces MixNAM, an extension of Neural Additive Models (NAMs) designed to enhance both accuracy and interpretability. MixNAM incorporates a mixture of experts, each capturing different aspects of feature variability, to address the limitations of traditional NAMs, which struggle to represent complex data patterns. Experiments show that MixNAM improves accuracy over traditional additive models while achieving performance comparable to black-box models, achieving a balance between interpretability and predictive power.

**Strengths:**

- The paper introduce MoE within the NAM framework to overcoming the limitations of traditional additive models. This architecture allows MixNAM to capture data complexity more effectively than NAMs.
- The paper includes experiments across various datasets, including both real-world and simulation data, to demonstrate MixNAM's improved accuracy and interpretability compared to both traditional additive models and more complex black-box approaches.

**Weaknesses:**

- Although MixNAM achieves interpretability with improved accuracy, it relies on a dynamic routing mechanism and multiple experts, which might increase computational requirements.  It would be better to include analysis of computational cost and assess the tradeoff of the computational cost and the increased accuracy.
- The paper primarily focuses on tabular data, which raises questions about the generalizability of the framework’s effectiveness to other domains, such as image or text data.

**Questions:**

- For datasets with highly sparse features, does the routing mechanism maintain its efficiency, or does it introduce sparsity issues that affect performance?
- The simulation study includes only two random variables. Could the study include more experiments on high dimensional variables to better demonstrate MixNAM's effectiveness?
- How does MixNAM handle extreme cases of feature variability, where the impact of a feature varies widely across different samples?

---

> ### Author Response · Authors · 2024-11-20
> **Response to Reviewer NxyM**
>
> Thank you for providing thoughtful and constructive feedback. Following your suggestions, we conducted additional simulation studies and added the results to the updated manuscript. Here are our detailed responses to your questions.
>
> ### Weaknesses:
>
> > Although MixNAM achieves interpretability with improved accuracy, it relies on a dynamic routing mechanism and multiple experts, which might increase computational requirements. It would be better to include analysis of computational cost and assess the tradeoff of the computational cost and the increased accuracy.
>
> Thank you for the suggestion! We have analyzed the additional computational cost introduced by the mixture of experts in Appendix J (previously Appendix I). Our analysis shows that there is a quadratic cost increase with respect to the number of features in MixNAM due to the feature interaction modeling within the routing system, which leads to the observed improvement in accuracy.
>
> > The paper primarily focuses on tabular data, which raises questions about the generalizability of the framework’s effectiveness to other domains, such as image or text data.
>
> In the existing literature on additive models, evaluations primarily focus on tabular data, while applications to other modalities, such as text or images, are rare [1,2,3]. This is because the core capability of additive models lies in interpreting features through shape plots, which illustrate how predictions change as feature values monotonically increase or decrease. Additive models are difficult to evaluate on raw text or image data, where features and their monotonic relationships are challenging to define.
>
> For example, NBM [2] tested its performance on image data by using concept bottleneck models to preprocess images into tabular data. However, the effectiveness of the overall system was limited by the quality of the extracted concepts, which may not fully capture the information in the original images [4,5]. Therefore, we follow the mainstream in additive model research and evaluate MixNAM using tabular data. We have discussed the potential generalization of MixNAM to other modalities as a future direction in Appendix L (previously Appendix K).
>
> [1] Agarwal R, et al. Neural additive models: Interpretable machine learning with neural nets. NeurIPS 2021.
>
> [2] Radenovic F, et al. Neural basis models for interpretability. NeurIPS 2022.
>
> [3] Chang CH, et al. NODE-GAM: Neural Generalized Additive Model for Interpretable Deep Learning. ICLR 2022.
>
> [4] Koh PW, et al. Concept bottleneck models. ICML 2020.
>
> [5] Margeloiu A, et al. Do concept bottleneck models learn as intended?. Arxiv 2021.
>
> ### Questions:
>
> > For datasets with highly sparse features, does the routing mechanism maintain its efficiency, or does it introduce sparsity issues that affect performance?
>
> We have added a new simulation study to investigate MixNAM’s performance on data with sparse features. Using the original simulation settings, we generated multimodal data with two features $x_1$ and $x_2$. In this new study, we manually controlled the proportion of data points with $x_2=1$ to represent 25%, 5%, and 1% of the dataset. The visual results for NAM and MixNAM are shown in Figure 6 in the updated manuscript.
>
> The results demonstrate that NAM tends to focus on the majority group when the signal for $x_2=1$ is sparse. In contrast, MixNAM successfully identifies both modalities in the data distribution and learns the sparse distribution better with an increasing number of experts.
>
> > The simulation study includes only two random variables. Could the study include more experiments on high dimensional variables to better demonstrate MixNAM's effectiveness?
>
> > How does MixNAM handle extreme cases of feature variability, where the impact of a feature varies widely across different samples?
>
> We added a new simulation study to explore how MixNAM performs with high-dimensional variables and features with extreme variability in their output contributions. To investigate this, we simulated a data distribution using the following pattern:
> $$y=\varepsilon + \sum_{i=2}^{NC+1}\frac{T}{NC}x_i\sin(4\pi x_1),$$
> where the feature $x_1$ is sampled from $U(0, 1)$ and $x_2,\cdots,x_{NC+1}$ are the $NC$ categorical features sampled from \{-1, +1\} with equal probability. $T$ is a scaling factor that amplifies variability across different modalities. For this study, we set $T = 64$ and evaluated both NAM and MixNAM across scenarios with $NC = 1, 2, 4, 8, 16$.
>
> As illustrated in Figure 7 of the updated manuscript, the results demonstrate that MixNAM effectively captures the multimodal contributions of $x_1$ to the output, showing consistent performance as the number of features and modalities increases. In contrast, NAM struggles to account for multimodality due to its inherent limitation of feature additivity. The results highlight the robustness and scalability of MixNAM in handling multimodal data with high variability.

---

> ### Comment · Reviewer_NxyM · 2024-12-03
>
> Thank the authors for their detailed response. My questions have been mostly solved. I will keep my original positive score.

---

### Official Review · Reviewer_a4Jz · 2024-11-04

**Soundness:** 2
**Presentation:** 3
**Contribution:** 2
**Rating:** 3
**Confidence:** 4

**Summary:**

Additive models like Neural Additive Models (NAMs) are valued for their transparency, clearly showing how individual features impact outcomes. However, their reliance on point estimates and additive structure limits their ability to capture complex, variable feature influences in real-world data. To address these limitations, MixNAM is introduced as a framework that enriches NAMs through a mixture of experts, each capturing different aspects of feature variability. This approach allows MixNAM to model diverse feature contributions, and interactions, and allow distribution estimations. Empirical results show that MixNAM not only outperforms traditional additive models but also approaches the performance of complex black-box methods while providing detailed feature attributions. Its flexible configuration further allows for balancing accuracy and interpretability, adapting to various data scenarios effectively.

**Strengths:**

- Overall, the paper is clear and well-written.
- The paper attempts to contribute to the important and relevant topic of uncertainty quantification in neural additive models with mixtures of experts
- The proposed method is evaluated on both simulated and real-world data, using multiple criteria, including model additivity, bound tightness, and prediction accuracy.

**Weaknesses:**

* The discussion on related work lacks clarity. For instance, the authors state that "these approaches are still limited by their inevitable assumptions of prior distributions and the additive nature of the entire models, which restrict their ability to accurately reflect complex underlying distributions and interactions." However, it is unclear what is meant by "prior distributions." Why can’t these models capture interactions effectively? Why are they considered less flexible? Additionally, the authors mention that "these models generally rely on a simplistic assumption about output distributions." More specific details are needed to understand the limitations that the proposed approach aims to address.

* The type of uncertainty or variability captured by the proposed method is not clearly defined. The authors use various terms, such as "different aspect of the variability," "more comprehensive insight into the output distribution," "captures a broad range of possible outcomes," "detailed variance," and "more comprehensive insight into the output distribution." However, this language lacks rigor and does not clearly communicate the specific type of uncertainty intended to be captured. Furthermore, in a regression context where \(L\) represents the MSE, it is unclear how the proposed method would capture "uncertainty" by minimizing MSE (instead of proper scoring rules).

* Although the authors consider different values for \(K\) and \(C\) in Table 8 in the appendix, the roles of these hyperparameters are not well explained. The results appear relatively consistent, and the impact of using a single expert is missing from the analysis. Additionally, there is no study of the values for \(\gamma\) and \(\lambda\) in equation (8), particularly with extreme values (gamma = 0).

* As demonstrated by the authors in Appendix E, the proposed method can essentially be viewed as a generalized additive model with specific normalization. The added complexity in the approach is not well justified when compared to existing methods.

* Only six relatively "old" datasets are used in this study. A broader selection of available tabular datasets would provide a stronger and more comprehensive evaluation. Refer to Léo Grinsztajn et al. “Why do tree-based models still outperform deep learning on tabular data?” (July 2022) and Pieter Gijsbers et al. “An Open Source AutoML Benchmark” (July 2019) for relevant data sources.

* The authors write, "The bounds represent the maximum and minimum potential outputs for a feature." It would be helpful to clarify what is meant by "possible" in this context. Possible according to what criteria? These bounds are directly affected by the number of experts. How is that important?


Additional Comments

* In expression (12), the denominator equals zero.
* In Figure 4, why is NAM unable to capture multimodality? Were proper hyperparameters used?
* Please clearly indicate what the values after the +/- symbol represent (standard errors?).
* Label the y-axis in Figure 3 for clarity.

**Questions:**

See weaknesses.

---

> ### Author Response · Authors · 2024-11-20
> **Response to Reviewer a4Jz (Part 1)**
>
> We appreciate your detailed and valuable comments and would like to take this opportunity to clarify our contribution and address the points you have raised. Below are our responses to your questions.
>
> ### Weaknesses:
> > The discussion on related work lacks clarity [...]
>
> By "prior distributions", we mean that existing NAM research with uncertainty estimation introduce explicit assumptions about the distributions they model. For example, [1] "impose a zero-mean Gaussian prior distribution over the parameters of each feature network". NAMLSS [2] is tailored to model a pre-determined distribution, "e.g. a normal distribution".
>
> Traditional additive models cannot effectively capture feature interactions due to their architecture:
> $$\hat{y}=w_0+f_1(x_1)+\cdots+f_n(x_i),$$
> where each feature's contribution is modeled independently. While they incorporate prior distributions to increase uncertainty and complexity of the model prediction, their additive structure prevents interactions from being represented.
>
> By "simplistic assumptions about output distributions," we refer to the additive constraints inherent in these architectures mentioned above. Our work aims to exceed these additive constraints, introducing a framework that models complex distributions while maintaining interpretability, balancing the accuracy and interpretability for real-world data analysis.
>
> [1] Improving neural additive models with bayesian principles. ICML 2024.
>
> [2] Neural additive models for location scale and shape: A framework for interpretable neural regression beyond the mean. AISTATS 2024.
>
> > The type of uncertainty or variability captured by the proposed method is not clearly defined [...]
>
> Thank you for the thoughtful comment! We agree it is valuable to formally define the "variability" in a mathematical way, which will help clarify and highlight the contribution of this work. Our proposed MixNAM seeks to capture variability **not in the randomness of predictions**, but **in the way other features influence the contribution of a specific feature $x_i$ to the final output $\hat{𝑦}$**. This approach allows for the modeling of feature interactions and goes beyond traditional additive models, which inherently assume no interactions between features.
>
> Consider the mapping from the input features $x_1,\cdots,x_n$ to the predicted output:
>
> $$\hat{y} = F(x_1, x_2, \cdots, x_n),$$
>
> where $F$ represents the underlying predictive function. For a fixed value of $x_i=a$ the variability of the contribution of $x_i$ to $\hat{y}$ can be formally defined as:
>
> $$variability_{x_i=a} = Var_{x_1,\cdots,x_n}[F(x_1, x_2, \cdots, x_n|x_i=a) - E_{x_i}(F(x_1, x_2, \cdots, x_n))].$$
>
> This definition measures how the contributions of $x_i=a$ deviate due to interactions with other features, reflecting variability caused by feature dependencies.
>
> For traditional additive models, the variability defined above reduces to zero because the contributions of each feature are independent and do not interact. In models where interactions exist, this term captures the extent to which other features $x_j (j\neq i)$ influence the contribution of $x_i$.
>
> For example, in our simulated unimodal data $y=\sin(4\pi x_1)+x_2$, the variability of $x_1$ at $x_1=a$ is:
>
> $$Var_{x_2}[\sin(4\pi a)+x_2-E_{x_1}(\sin(4\pi x_1))-x_2] = Var_{x_2}[\sin(4\pi a)]=0.$$
>
> For the multimodal data where $y=x_2\sin(4\pi x_1)+x_2$, the variability of $x_1$ at $x_1=a$ is:
>
> $$Var_{x_2}[x_2\sin(4\pi a)+x_2-E_{x_1}(x_2\sin(4\pi x_1))-x_2] = Var_{x_2}[x_2\sin(4\pi a)]=\sin^2(4\pi a)$$
>
> By modeling such uncertainty/variability with MoE, we are trying to close the gap between interpretable but poor-performing additive models and powerful but uninterpretable black-box models/true data distributions.

---

> ### Author Response · Authors · 2024-11-20
> **Response to Reviewer a4Jz (Part 2)**
>
> > [...] the roles of these hyperparameters are not well explained. The results appear relatively consistent, and the impact of using a single expert is missing from the analysis. Additionally, there is no study of the values for (\gamma) and (\lambda) in equation (8) [...]
>
> The roles of $C$ ("the total number of experts") and $K$ ("the number of activated experts") are clearly described in the main paper and in the appendix (lines 184, 213, 1004). These are well-established concepts in the Mixture of Experts (MoE) literature [1,2].
>
> We respectfully disagree with the conclusion that the results are "relatively consistent." As detailed in Table 8, configurations with optimal values of $K$ and $C$ significantly outperform settings with smaller numbers of activated experts (e.g., $K=2$). This demonstrates the importance of selecting appropriate values for these hyperparameters. Furthermore, using a single expert ($K=1$) corresponds to the behavior of traditional additive models, whose results are thoroughly presented and discussed in Table 1.
>
> Regarding $\lambda$, we have provided a detailed analysis of its effect on performance and interpretability in Tables 2 and 3. The results demonstrate how varying $\lambda$ allows us to balance accuracy and interpretability, with higher values emphasizing interpretability at the cost of performance.
>
> For $\gamma$, we opted not to focus on this parameter in our analysis, as it is not a novel component introduced by MixNAM. Instead, $\gamma$ is a well-established feature in prior additive model research [2,3]. It was included to ensure a fair comparison with existing methods rather than as a focal point of our study.
>
> [1] Outrageously large neural networks: The sparsely-gated mixture-of-experts layer. ICLR 2017.
>
> [2] Mixtral of experts. Arxiv 2024.
>
> [3] Neural Additive Models: Interpretable Machine Learning with Neural Nets. NeurIPS 2021.
>
> [4] Neural Basis Models for Interpretability. NeurIPS 2022.
>
> > As demonstrated by the authors in Appendix E, the proposed method can essentially be viewed as a generalized additive model with specific normalization. The added complexity in the approach is not well justified when compared to existing methods.
>
> Our analysis in Appendix E demonstrates that the relevance estimation in the routing mechanism of MixNAM, specifically described by Formula (6), follows a normalized Generalized Additive Model (GAM). However, it is important to clarify that this does not imply that MixNAM as a whole is equivalent to a normalized GAM. MixNAM extends beyond the scope of traditional additive models by incorporating a Mixture of Experts (MoE) framework, which dynamically captures feature interactions and variability through expert routing mechanisms.
>
> > Only six relatively "old" datasets are used in this study. A broader selection of available tabular datasets would provide a stronger and more comprehensive evaluation. Refer to Léo Grinsztajn et al. “Why do tree-based models still outperform deep learning on tabular data?” (July 2022) and Pieter Gijsbers et al. “An Open Source AutoML Benchmark” (July 2019) for relevant data sources.
>
> The datasets selected for this study are widely recognized benchmarks, encompassing a broad range of scenarios across regression and classification tasks. These datasets, including Housing, MIMIC-II, MIMIC-III, Income, Credit, and Year, provide diverse challenges in terms of complexity, feature types, and task objectives.
>
> While we acknowledge that additional datasets could be explored, we believe the results presented in Table 1 sufficiently demonstrate the effectiveness of MixNAM. The model consistently outperforms traditional additive models across all datasets, highlighting its capability to balance interpretability and performance.
>
> > The authors write, "The bounds represent the maximum and minimum potential outputs for a feature." It would be helpful to clarify what is meant by "possible" in this context. Possible according to what criteria? These bounds are directly affected by the number of experts. How is that important?
>
> The term "possible" in this context refers to the range of potential contributions of a given feature value $x_i = a$ to the final output. Specifically, this contribution is defined as:
> $$F(x_1, x_2, \cdots, x_n|x_i=a) - E_{x_i}(F(x_1, x_2, \cdots, x_n)),$$ which depends on the values of other features $x_j$ ($j \neq i$). The bounds are determined by the maximum and minimum values of this contribution term over all possible configurations of the other features. These bounds provide a rigorous characterization of the variability in feature contributions.
>
> It is important to note that the number of experts does not affect the bound estimation itself but influences the precision of the output predictions within the determined bounds. By using multiple experts, MixNAM captures a more nuanced representation of variability within the range defined by the bounds.

---

> ### Author Response · Authors · 2024-11-20
> **Response to Reviewer a4Jz (Part 3)**
>
> ### Additional Comments
>
> > In expression (12), the denominator equals zero.
>
> Thanks for pointing it out. The denominator should be `upper(o_i|x_i)-lower(o_i|x_i)`.
>
> > In Figure 4, why is NAM unable to capture multimodality? Were proper hyperparameters used?
>
> Yes, we tuned the hyperparameters of NAM to optimize its ability to fit multimodal data. However, NAM is inherently unable to capture multimodality because it provides only a single deterministic output value for each feature value. In contrast, the true underlying distribution for the multimodal data can yield multiple possible outputs for the same feature value.
>
> > Please clearly indicate what the values after the +/- symbol represent (standard errors?).
>
> Yes, the values after the "+/-" symbol represent the standard deviations of the metric scores across different runs. Following previous research, we tested models on the Housing and Year datasets using 10 different random seeds. For other datasets, evaluations were conducted using five-fold cross-validation. More details about the implementation settings can be found in Appendix B.
>
> > Label the y-axis in Figure 3 for clarity.
>
> The y-axis represents the contribution of each feature to the final prediction for a given instance. We have updated the manuscript to explicitly label the y-axis as "Output Contribution" for clarity.

---

### Author Response · Authors · 2024-11-22
**Summary Response to All Reviewers**

Dear Reviewers,

We sincerely appreciate your thoughtful reviews and constructive feedback on our manuscript. We have carefully addressed your comments and made corresponding revisions to the paper. Below, we summarize the major concerns raised and how they have been addressed in the updated manuscript:

- **Motivation of MixNAM** (Reviewers a4Jz, xFNJ): As described in the Introduction, the primary motivation for this work arises from the limitations of existing additive models, which often produce less accurate predictions than black-box models. Traditional additive models are inherently unable to capture multimodality, as they provide only a single deterministic output value for each input feature value. This limitation is clearly demonstrated through the simulation studies presented in Section 4.4.
- **Mathematical definition of variability captured by MixNAM** (Reviewer a4Jz, xFNJ): Beyond the conceptual explanation provided in the Introduction, we have added a formal mathematical definition of "variability" in Appendix E.2. This definition measures how other features influence the contribution of a given feature $x_i$ to the final output $\hat{𝑦}$. By modeling such variability with MixNAM, we are trying to bridge the gap between interpretable but lower-performing additive models the more powerful but opaque black-box models.
- **Discussion of post-hoc feature attribution methods** (Reviewer MzpU): We have included a new paragraph in Appendix D to highlight the uniqueness of MixNAM compared to post-hoc feature attribution methods such as LIME and SHAP. Unlike these methods, MixNAM provides a transparent, global understanding of feature contributions directly through its model design, avoiding the limitations of approximations inherent in post-hoc approaches.
- **Additional experiments on highly sparse data** (Reviewer NxyM): We have conducted an additional simulation study, the results of which are presented in Appendix E.1. These experiments demonstrate MixNAM’s ability to identify multimodality even in extreme cases of high sparsity.
- **Additional experiments on larger-scale multimodal data** (Reviewers NxyM, xFNJ): We have also included the results of another simulated study in Appendix E.2. This study showcases MixNAM’s capability to handle larger-scale datasets with pronounced multimodality and variability in feature contributions.

We hope these revisions address your concerns and enhance the overall quality of the paper. We would also appreciate if you could consider updating your scores based on the clarified contributions and the revised manuscript. Please let us know if you have any further concerns or suggestions.

Thank you once again for your time, effort, and valuable feedback!

Warm regards,
Authors

---

### Meta-Review · Area_Chair_YVJj · 2024-12-23

**Metareview:**

Summarization

The paper addresses the limitations of Neural Additive Models (NAMs), which, while interpretable and simple, fail to capture the complex relationships in data. Specifically, NAMs assume that the influence of a feature is uniform across all instances, ignoring variations in importance caused by feature values or contextual differences. To solve this, the paper proposes a Mixed Neural Additive Model that integrates multiple feature-specific experts. Each expert captures a distinct aspect of feature influence, and their outputs are dynamically combined using a routing mechanism. This design improves flexibility while retaining interpretability.

Experiments show that the proposed method outperforms traditional additive models and achieves results comparable to non-additive models. Additionally, it provides visualizations of feature contribution distributions. The paper said, that unlike existing solutions that rely on restrictive assumptions or compromise the additive structure, the proposed method gives a better balance between performance and interoperability.

Strengths

The main strengths have been twofolds 1) the proposed method is novel to capture the feature variety overcoming the shortage of traditional neural additive model; 2) the demonstrated experimental results show the effectiveness of the proposed method especially in balancing the interpretability and performance.

Weaknesses

Overall, I think there are three major concerns of the paper 1) the paper did not demonstrate sufficient technical novelty. As pointed out by two of the reviewers who gave 3, the paper seems to be a combination of the “mix of experts” and the “natural additive model”, with a weighted combination of experts. 2)The presentation is vague, especially on some of the key aspects, such as the motivation of the paper, and the contribution of the paper. Some terms are not mathematically well-defined. 3) More datasets and comparisons are required to fully verify the effectiveness of the proposed method.

Recommendation

Overall, while I agree with reviewers that the paper explores an important direction to consider the feature correlations and balance between interpretability and performance – the current version does not fully satisfy the standard to get published as concerns raised by reviewers regarding the technical contribution, empirical evaluation and clarity in terminology and contribution have not been fully addressed.

**Additional Comments On Reviewer Discussion:**

The paper has got 4 reviews with diverse ratings (3, 3, 6, 6). Two of them give a six (marginally above the threshold) – they have expressed satisfaction with the response of authors but did not increase the score (actually they have both explicitly indicated they will keep the score), which shows that although the questions they raised have been solved in some satisfiable extent to them, they still thought the paper is not good enough to be given a fully accept. The other two reviewers only give a rate of 3 (clear reject). It is a pity these two reviewers did not reply or involved in discussions. Their major concerns are listed in the weakness part above. From my own judgement of reading the rebuttal, the rebuttal argues that the purpose of the paper is not as limited as the reviewer stated – but that does not change the nature of the technical contribution. To address the question of novelty, arguments such as the challenge of directly applying MoE are required – which is missing from the rebuttal.  The reviewers have asked for additional experiments such as on large-scale real datasets or more datasets in existing literature, while the rebuttal only provided additional results on simulated datasets – which is understandable during the limited-time rebuttal phase, but the concern is not fully addressed.  Regarding the vague definitions of the terms motivation or contribution, the rebuttal has provided their mathematical formulation, while their mathematical formulations have not been clearly shown to be improved by the proposed method.

---

### Decision · Program_Chairs · 2025-01-22

Reject